# Fine-tuning of β-catenin in mouse thymic epithelial cells is required for postnatal T-cell development

**Sayumi Fujimori[1,2]\*, Izumi Ohigashi[1], Hayato Abe[3], Yosuke Matsushita[4], Toyomasa Katagiri[4], Makoto M Taketo[5], Yousuke Takahama[6]\*, Shinji Takada[2,7,8]\***

[1]Division of Experimental Immunology, Institute of Advanced Medical Sciences, Tokushima University, Tokushima, Japan; [2]National Institute for Basic Biology, National Institutes of Natural Sciences, Okazaki, Japan; [3]Student Laboratory, School of Medicine, Tokushima University, Tokushima, Japan; [4]Division of Genome Medicine, Institute of Advanced Medical Sciences, Tokushima University, Tokushima, Japan; [5]Institute for Advancement of Clinical and Translational Science, Kyoto University Hospital, Kyoto, Japan; [6]Experimental Immunology Branch, National Cancer Institute, National Institutes of Health, Bethesda, United States; [7]Exploratory Research Center on Life and Living Systems, National Institutes of Natural Sciences, Okazaki, Japan; [8]Department of Basic Biology in the School of Life Science, The Graduate University for Advanced Studies (SOKENDAI), Okazaki, Japan

**\*For correspondence:**
sfujimori@genome.tokushima-u.ac.jp (SF);
yousuke.takahama@nih.gov (YT);
stakada@nibb.ac.jp (ST)

**Competing interest:** The authors declare that no competing interests exist.

**Abstract** In the thymus, the thymic epithelium provides a microenvironment essential for the development of functionally competent and self-tolerant T cells. Previous findings showed that modulation of Wnt/β-catenin signaling in mouse thymic epithelial cells (TECs) disrupts embryonic thymus organogenesis. However, the role of β-catenin in TECs for postnatal T-cell development remains to be elucidated. Here, we analyzed gain-of-function (GOF) and loss-of-function (LOF) of β-catenin highly specific in mouse TECs. We found that GOF of β-catenin in TECs results in severe thymic dysplasia and T-cell deficiency beginning from the embryonic period. By contrast, LOF of β-catenin in TECs reduces the number of cortical TECs and thymocytes modestly and only postnatally. These results indicate that fine-tuning of β-catenin expression within a permissive range is required for TECs to generate an optimal microenvironment to support postnatal T-cell development.

## Editor's evaluation

This paper is of interest to scientists within the field of thymus development and function. The work presented builds upon previous studies that have shown that alterations of Wnt/β-catenin signaling in thymic epithelial cells (TEC) impact the normal development and or maintenance of thymic epithelial microenvironment critical for the proper development and selection of functional self-tolerant T cell repertoire. The surprise that a TEC specific loss of function of β-catenin only showed a rather minor phenotype is interesting, as are the findings that gain of function of β-catenin leads to TEC differentiation towards a keratinocyte-like lineage outcome. The author's claims are well supported by the data presented and will be of great interest to scientists and clinicians interested in understanding the signaling pathways important in thymic maintenance, as well as the development of strategies to counteract thymic involution in the aging population and cancer patients.

## Introduction

The thymus is an organ where lymphocyte progenitors differentiate into functionally competent T cells through interactions primarily with cortical thymic epithelial cells (cTECs) and medullary thymic epithelial cells (mTECs), which construct a three-dimensional epithelial network of the thymus and facilitate the development and selection of developing T cells. To establish a functional thymic microenvironment, the thymus organogenesis is tightly regulated by the interaction of thymic epithelial cells (TECs) with neighboring cells, such as mesenchymal cells and hematopoietic cells, through various molecules including morphogens, growth factors, cytokines, and chemokines, during the thymus organogenesis that is initiated around embryonic day 11 (E11) in mouse (*Blackburn and Manley, 2004*; *Gordon and Manley, 2011*; *Takahama et al., 2017*).

The Wnt/β-catenin signaling pathway has been implicated in thymus development following the discovery of its core components, T-cell factor (TCF)/lymphoid enhancing factor (LEF) family transcription factors, in developing thymocytes (*Oosterwegel et al., 1991*; *Travis et al., 1991*). Wnt proteins activate the canonical signaling pathway through interaction with their receptors of the Frizzled family and coreceptors of the LRP5/6 family. This interaction leads to the inhibition of the β-catenin destruction complex, which is composed of adenomatous polyposis coli (APC), axin, casein kinase 1α (CK1α), and glycogen synthase kinase 3β (GSK3β) proteins, thereby resulting in the stabilization of cytosolic β-catenin, its translocation to the nucleus, and enhancement of transcription together with the TCF/LEF family transcription factors (*Nusse and Clevers, 2017*).

Previous systemic and conditional gene manipulation studies in mouse have demonstrated the importance of Wnt/β-catenin signaling in early thymus development. The systemic deletion of Wnt4, which is abundant in embryonic TECs (*Pongracz et al., 2003*), reduces the number of TECs as well as thymocytes (*Heinonen et al., 2011*). Keratin 5 (*Krt5*) promoter-driven inhibition of Wnt/β-catenin signaling by the overexpression of Wnt inhibitor Dickkopf1 (DKK1) or by the deficiency of β-catenin reduces the number of TECs including K5$^+$K8$^+$ immature TECs (*Osada et al., 2010*; *Liang et al., 2013*). Similarly, a reduction in the number of TECs is observed in perinatal mice in which β-catenin is deleted using Foxn1-Cre, which targets TECs expressing transcription factor Forkhead box N1 (Foxn1) (*Swann et al., 2017*). However, the Foxn1-Cre-mediated β-catenin deletion causes neonatal lethality due to side effects in the skin epidermis, so that the postnatal effects of β-catenin deficiency in TECs remain unknown. On the other hand, overactivation of Wnt/β-catenin signaling by the deletion of transmembrane protein Kremen 1, which binds to DKK1 and potentiates Wnt signaling inhibition, results in the increase in the number of K5$^+$K8$^+$ immature TECs and the disorganization of the TEC network by the decrease in the frequency of mature cTECs and mTECs (*Osada et al., 2006*). Keratin 14 (*Krt14*) promoter-driven loss of APC disorganizes the thymus and reduces its size (*Kuraguchi et al., 2006*). The overexpression of β-catenin using Foxn1-Cre causes a more severe defect in embryonic thymus development, which is accompanied by the failure of separation of the thymic primordium from the parathyroid primordium (*Zuklys et al., 2009*; *Swann et al., 2017*). These results indicate that Wnt/β-catenin signaling is critical for embryonic TEC development. However, the role of Wnt/β-catenin signaling in TECs for thymus development and function during the postnatal period remains unclear because previous studies relied on mutant mice that exhibit perinatal lethality due to severe effects on various cell types including the skin epidermis.

In this study, we assessed the role of β-catenin in TECs by employing β5t-iCre, which was successfully engineered to target TECs in a highly specific manner without side effects in other cells including the skin epithelium (*Ohigashi et al., 2013*). Thymus-specific proteasome subunit β5t is specifically expressed in cTECs in a Foxn1-dependent manner, and is important for the generation of functionally competent CD8$^+$ T cells (*Murata et al., 2007*; *Ohigashi et al., 2021*). β5t in postnatal mice is exclusively expressed by cTECs but not mTECs (*Ripen et al., 2011*; *Uddin et al., 2017*); however, its transcription is also detectable in embryonic TECs that differentiate into cTECs and mTECs (*Ohigashi et al., 2013*; *Ohigashi et al., 2015*). By taking advantage of the highly specific Cre activity only in TECs and not in skin epidermal cells in β5t-iCre mice, here we engineered gain-of-function (GOF) and loss-of-function (LOF) of β-catenin specifically in TECs and addressed the contribution of the Wnt/β-catenin signaling pathway in TECs, focusing on the effects of thymus function on T-cell development during the postnatal period.

# Results

## GOF of β-catenin in TECs causes thymic dysplasia

To address the potential function of β-catenin in TECs, we generated mice in which GOF of β-catenin was achieved specifically in TECs by intercrossing β5t-iCre mice (*Ohigashi et al., 2013*) with *Ctnnb1* exon 3 floxed mice, in which the conditional deletion of exon 3 led to stabilization of β-catenin and constitutive activation of β-catenin signaling (*Harada et al., 1999*). Contrary to previous studies demonstrating that the Foxn1-Cre-mediated GOF of β-catenin perturbed the migration of the thymic primordium from the pharyngeal region into the thoracic cavity during the embryogenesis (*Zuklys et al., 2009*; *Swann et al., 2017*), the β5t-Cre-mediated GOF of β-catenin (β-cat GOF) showed no disruption of the thoracic migration of the embryonic thymus (*Figure 1A, B*). However, the thymus in these mice showed severe dysplasia by E15.5 (*Figure 1B, C*). The β-cat GOF embryos appeared normal in size (*Figure 1B*) and exhibited undisturbed limb development (data not shown), indicating that the thymic dysplasia in β-cat GOF mice occurred independent of systemic retardation in embryogenesis. Fluorescence detection of Cre-mediated recombination by further crossing to R26R-tdTomato reporter mice (*Madisen et al., 2010*) revealed that β5t-Cre-mediated tdTomato-expressing cells were specifically detected in the thymus in the thoracic cavity in both control and β-cat GOF mice at E15.5, further confirming that TECs successfully migrated into the thoracic cavity in β-cat GOF mice (*Figure 1B*). Unlike control mice in which tdTomato-labeled cells distributed uniformly throughout the thymus, tdTomato-labeled cells in β-cat GOF mice densely distributed in concentric mid-inner layers, excluding the central core, in the thymus (*Figure 1B*). Morphological alteration of the thymus was apparent in sagittal sections of E15.5 β-cat GOF embryos by hematoxylin and eosin staining and by the detection of keratin 5 (K5) and keratin 8 (K8) expression, in which a multilayered concentric structure of epithelial cells surrounding the central core that mainly consisted of nonepithelial structures sparsely containing erythrocytes was observed (*Figure 1C*, *Figure 1—figure supplement 1*). β-Catenin was abundant in the concentric layers of the thymus in E15.5 β-cat GOF mice, whereas β5t was undetectable throughout the E15.5 β-cat GOF thymus (*Figure 1C*), indicating that GOF of β-catenin in TECs diminishes β5t expression in TECs. Intracellular staining of β-catenin in CD45⁻EpCAM⁺ TECs isolated from E15.5 thymus revealed that the intracellular level of β-catenin in TECs from β-cat GOF mice was increased by 70% compared with that from control mice (*Figure 1D*). These results indicate that β5t-Cre-mediated GOF of β-catenin specifically in TECs causes severe thymic dysplasia by E15.5 without affecting thoracic migration of the thymic primordium.

## Defective thymus development by GOF of β-catenin in TECs

To examine how early the thymic abnormality was detectable in the embryos of β-cat GOF mice, we next performed histological analysis of the thymic primordium at early stages of fetal thymus development before E15.5. The expression of Foxn1, a transcription factor that characterizes the thymic epithelium-specified domain of the third pharyngeal pouch (*Nehls et al., 1994*; *Gordon et al., 2001*), was detected in the thymic primordium in both β-cat GOF and control embryos at E11.5, indicating that the initiation of the earliest thymus organogenesis is undisturbed in β-cat GOF mice (*Figure 2A*, *Figure 2—figure supplement 1A*). At E12.5, when the expression of β5t becomes detectable in the thymus (*Ripen et al., 2011*), the colonization of CD45⁺ lymphocytes in Foxn1-expressing thymic primordium was observed in both β-cat GOF and control embryos (*Figure 2A*), further supporting that early thymus organogenesis is undisturbed in β-cat GOF mice. However, the thymus morphology was markedly altered in β-cat GOF embryos at E13.5, namely, the thymic primordium appeared spherical in β-cat GOF embryos and ovoid in control embryos (*Figure 2A*). At this stage in β-cat GOF embryos, Foxn1⁺ TECs and CD45⁺ lymphocytes accumulated preferentially in the anterior area of the thymic primordium (*Figure 2A*). By E15.5, Foxn1 expression became undetectable and the number of CD45⁺ lymphocytes was markedly reduced in the dysplastic thymus in β-cat GOF embryos (*Figure 2A*).

Quantitative RT-PCR analysis of CD45⁻EpCAM⁺ TECs isolated from E15.5 thymus established that the expression of Wnt/β-catenin target genes, including *Axin2*, *Dkk1*, and *Msx1*, was upregulated in β-cat GOF TECs (*Figure 2B*, *Figure 2—figure supplement 1B*). By contrast, the expression of genes functionally relevant to TECs, such as *Foxn1*, *Psmb11* (encoding β5t protein), and *Ccl25* (encoding chemokine CCL25 protein that attracts lymphocyte progenitors to the thymic primordium; *Liu et al., 2006*), was significantly reduced in β-cat GOF TECs, consistent with the results of immunohistochemical analysis. The initial attraction of lymphoid progenitors to the

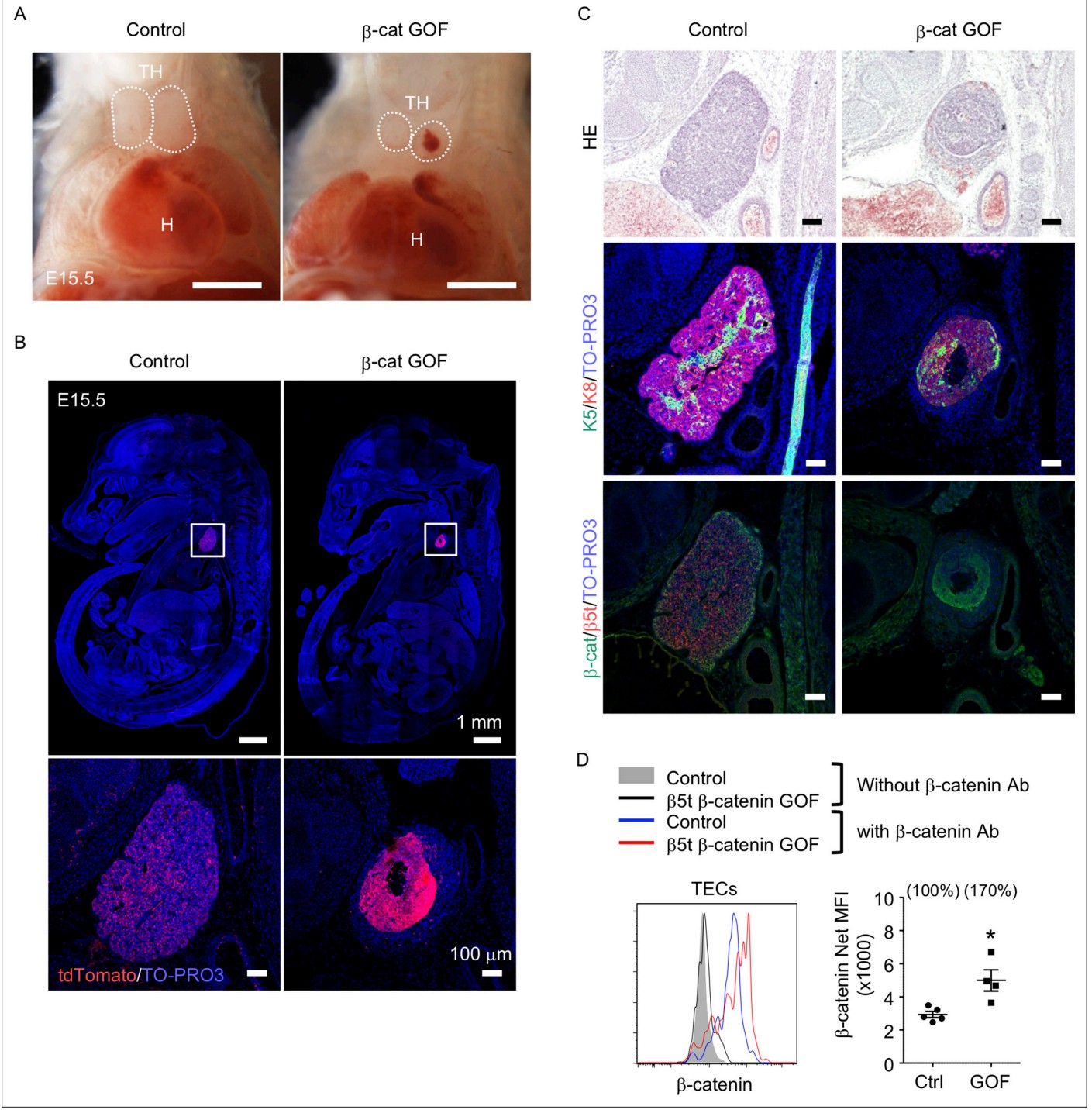

**Figure 1.** Thymic dysplasia caused by the overexpression of β-catenin in thymic epithelial cells (TECs). (**A**) Thoracic cavity of control mice and β-cat gain-of-function (GOF) mice at E15.5. Dotted lines show the outline of the thymic primordium. In many cases, blood clots were observed in the central core of the thymic primordium in β-cat GOF mice. Representative data from three independent experiments are shown. TH: thymus, H: heart. Bar: 1 mm. (**B**) Labeling of β5t-iCre-activated cell progenies with tdTomato fluorescence in the thymus of control mice and β-cat GOF mice at E15.5. Mouse sections were nuclear counterstained with TO-PRO3. Bottom panels are magnifications of white-boxed areas in the top panels. Representative data from three independent experiments are shown. Bars: indicated in figures. (**C**) Hematoxylin and eosin staining (top) and immunofluorescence staining for K5 and K8 (middle) and β-catenin and β5t (bottom) on sagittal sections of thymic primordium in control mice and β-cat GOF mice at E15.5. Representative results from three independent experiments are shown. Bar: 100 µm. (**D**) Intracellular staining of β-catenin in CD45⁻EpCAM⁺ TECs isolated from control mice and β-cat GOF mice at E15.5. Histograms show β-catenin expression in control TECs (blue line) and β-cat GOF TECs (red line). Shaded area and black line represent the fluorescence in the absence of anti-β-catenin antibody in control TECs and β-cat GOF TECs, respectively. Plots on the right show net

*Figure 1 continued on next page*

*Figure 1 continued*

median fluorescence intensity (MFI) values (means and standard error of the means [SEMs], *n* = 4–5). The numbers in parentheses indicate percentage of control value. *p < 0.05.

The online version of this article includes the following figure supplement(s) for figure 1:

**Figure supplement 1.** Immunofluorescence analysis of embryonic thymus.

thymic primordium is cooperatively regulated by CCL25 and CCL21 chemokines, the expression of which is dependent on Foxn1 and Glial Cells Missing Transcription Factor 2 (Gcm2), respectively (*Liu et al., 2006*). Given that the expression of *Ccl21a* was not altered in β-cat GOF TECs (data not shown), and given that β5t transcription for β5t-Cre-mediated β-catenin GOF is dependent on Foxn1 (*Žuklys et al., 2016*; *Uddin et al., 2017*), the overexpression of β-catenin in TECs primarily affected Foxn1-dependent *Ccl25* expression but not Gcm2-dependent *Ccl21a* expression. The expression of Delta-like (Dll) 4, a Foxn1-dependent Notch ligand essential for early T-cell development (*Hozumi et al., 2008*; *Koch et al., 2008*), was markedly reduced in β-cat GOF TECs (*Figure 2B*). Notably, the expression of involucrin (*Ivl*), loricrin (*Lor*), and desmoglein-1 alpha (*Dsg1a*) genes by the terminally differentiated keratinocytes (*Candi et al., 2005*) and the terminally differentiated subpopulation of mTECs (*Yano et al., 2008*; *Michel et al., 2017*) was upregulated in β-cat GOF TECs but not control TECs at E15.5 (*Figure 2B*). These results suggest that normal differentiation of TECs is disrupted in β-cat GOF embryos; instead, embryonic TECs exhibit aberrant differentiation into epithelial cells that partially share gene expression profiles with the terminally differentiated keratinocytes.

The number of CD45[+] thymocytes in E15.5 thymus was significantly reduced in β-cat GOF mice compared with control mice (*Figure 2C*). Flow cytometric analysis of CD45[+] thymocytes indicated that the majority of E15.5 thymocytes in β-cat GOF mice and control mice were equivalently CD4[−]CD8[−] double negative (DN) (data not shown). However, the DN thymocytes in β-cat GOF embryos were predominantly arrested at CD44[+]CD25[−] DN1 stage (*Figure 2D*). Indeed, the numbers of downstream thymocytes at DN2 (CD44[+]CD25[+]), DN3 (CD44[−]CD25[+]), and DN4 (CD44[−]CD25[−]) stages were dramatically reduced in β-cat GOF embryos compared with control embryos, whereas the number of DN1 thymocytes was comparable between β-cat GOF and control embryos (*Figure 2E*). These results indicate that embryonic thymus development is severely disturbed by β5t-Cre-mediated GOF of β-catenin specifically in TECs, which in turn results in early arrest of thymocyte development in embryonic thymus.

## GOF of β-catenin in TECs causes postnatal loss of T cells in secondary lymphoid organs

β-Cat GOF mice grew to adulthood with no gross abnormality in appearance and body weight (*Figure 3A, B*), whereas the thymic dysplasia persisted throughout the postnatal periods in β-cat GOF mice (*Figure 3C*). Flow cytometric analysis of cells from postnatal lymphoid organs showed that the frequency and the number of αβTCR-expressing T cells in the spleen and the inguinal lymph node (iLN) were significantly reduced in β-cat GOF adult mice (*Figure 4A, B*), indicating that αβ T cells are essentially lost in β-cat GOF mice. We also detected the reduction of γδ TCR-expressing T cells in β-cat GOF mice. Vγ5[+] γδT cells, which are generated in the embryonic thymus and localize to the epidermis to form dendritic epidermal T cells (DETCs) in adult mice (*Havran and Allison, 1988*), were severely reduced in frequency in the embryonic thymus and the postnatal skin of β-cat GOF mice (*Figure 4C, D*). Vγ4[+] and Vγ1[+] γδT cells, which are generated in the thymus during the perinatal period subsequently to the embryonic generation of Vγ5[+] γδT cells (*Pereira et al., 1995*), were similarly reduced in frequency in the iLN and the spleen of β-cat GOF adult mice (*Figure 4E*, data not shown). These results indicate that the thymus-dependent γδT cells, including Vγ5[+], Vγ4[+], and Vγ1[+] γδT cells, are essentially lost in β-cat GOF mice.

Taken together, our results substantiate that GOF of β-catenin in TECs causes severe thymic dysplasia that results in postnatal loss of thymus-derived αβ and γδ T cells in secondary lymphoid organs.

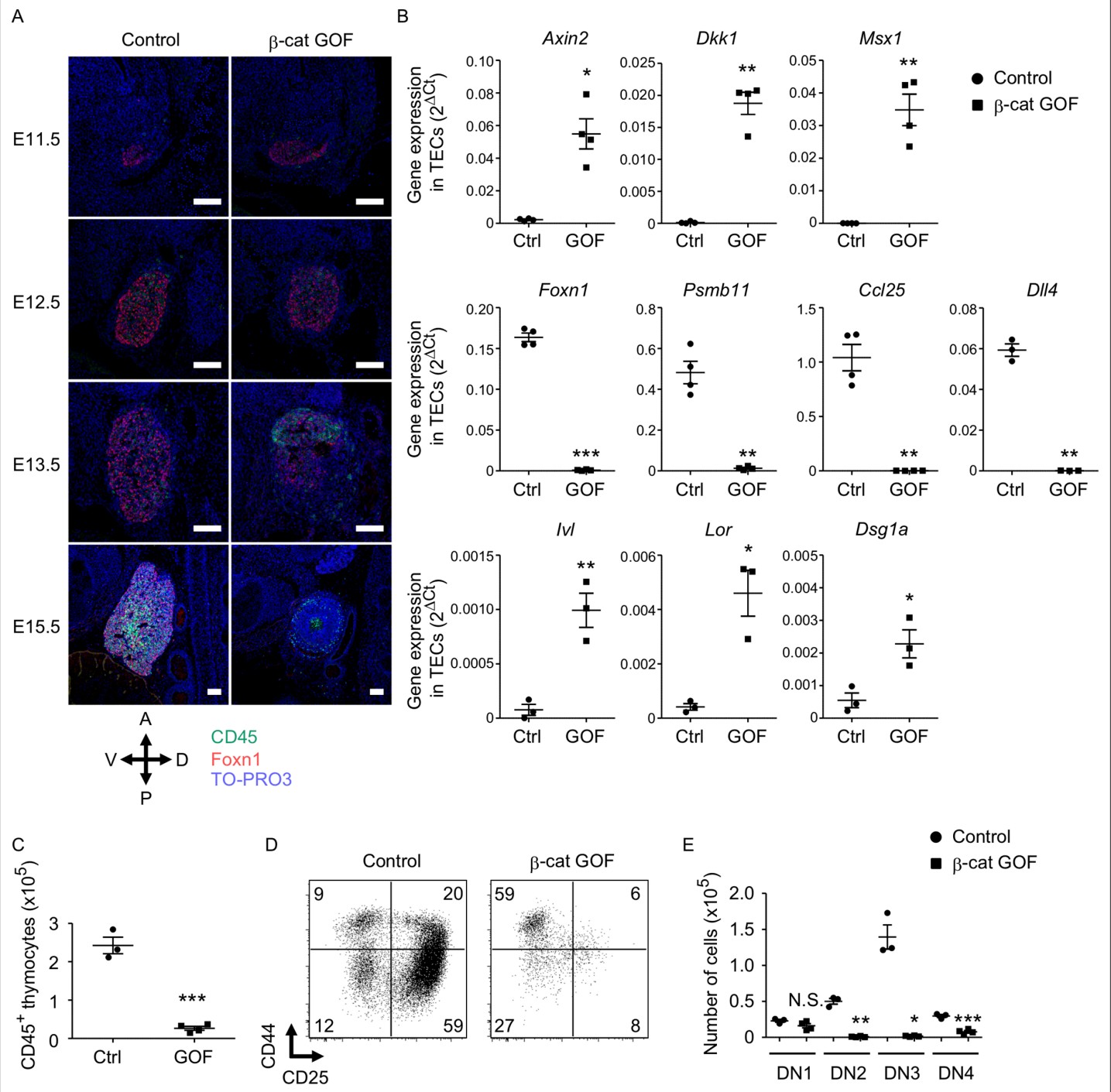

**Figure 2.** Defective thymus development in β-cat gain-of-function (GOF) embryos. (**A**) Immunofluorescence staining for CD45 and Foxn1 on sagittal sections of thymic primordium in control mice and β-cat GOF mice at E11.5–E15.5. The sections were nuclear counterstained with TO-PRO3. Anterior–posterior (A–P) and dorsal–ventral (D–V) orientations of the images are indicated. Representative data from three independent experiments are shown. Bar: 100 μm. (**B**) Quantitative RT-PCR analysis of mRNA expression levels (means and standard error of the means [SEMs], $n = 3$–4) of indicated genes relative to *Gapdh* levels in CD45⁻EpCAM⁺ thymic epithelial cells (TECs) isolated from the thymus of control mice and β-cat GOF mice at E15.5. (**C**) The numbers of CD45⁺ thymocytes were analyzed by flow cytometry. Plots show the numbers (means and SEMs, $n = 3$–4) of CD45⁺ thymocytes in the thymus of control mice and β-cat GOF mice at E15.5. (**D**) Flow cytometric analysis of double negative (DN) thymocytes from control mice and β-cat GOF mice at E15.5. Shown are profiles of CD44 and CD25 expression. The numbers in dot plots indicate the frequency of cells within indicated area. (**E**) Cell numbers (means and SEMs, $n = 3$–4) of indicated DN thymocyte subpopulations from control mice and β-cat GOF mice at E15.5 are plotted. $*p < 0.05$; $**p < 0.01$; $***p < 0.001$; N.S., not significant.

*Figure 2 continued on next page*

*Figure 2 continued*

The online version of this article includes the following figure supplement(s) for figure 2:

**Figure supplement 1.** Immunofluorescence analysis of embryonic thymus and purity of isolated TECs.

## LOF of β-catenin in TECs results in no apparent change in thymus development during the embryonic period

To further evaluate the role of β-catenin in TECs, LOF analysis of β-catenin was performed using *Ctnnb1* floxed allele targeted by the β5t-Cre-mediated recombination (β-cat LOF). In contrast to β-cat GOF mice, morphological abnormality in the thymus was not apparent in β-cat LOF embryos at E15.5 (*Figure 5A*). Immunohistofluorescence analysis confirmed the reduction of β-catenin in the thymus of β-cat LOF mice at E15.5, whereas the expression of Foxn1, K5, and K8 in E15.5 thymus was comparable between β-cat LOF mice and littermate control mice (*Figure 5B*). Flow cytometric analysis revealed that intracellular β-catenin level was reduced by 77% in CD45⁻EpCAM⁺ TECs of β-cat LOF mice at E15.5 (*Figure 5C*). However, the number of total thymic cells and the frequency and the number of total TECs and UEA1⁻Ly51⁺ cTECs were comparable between β-cat LOF and control mice

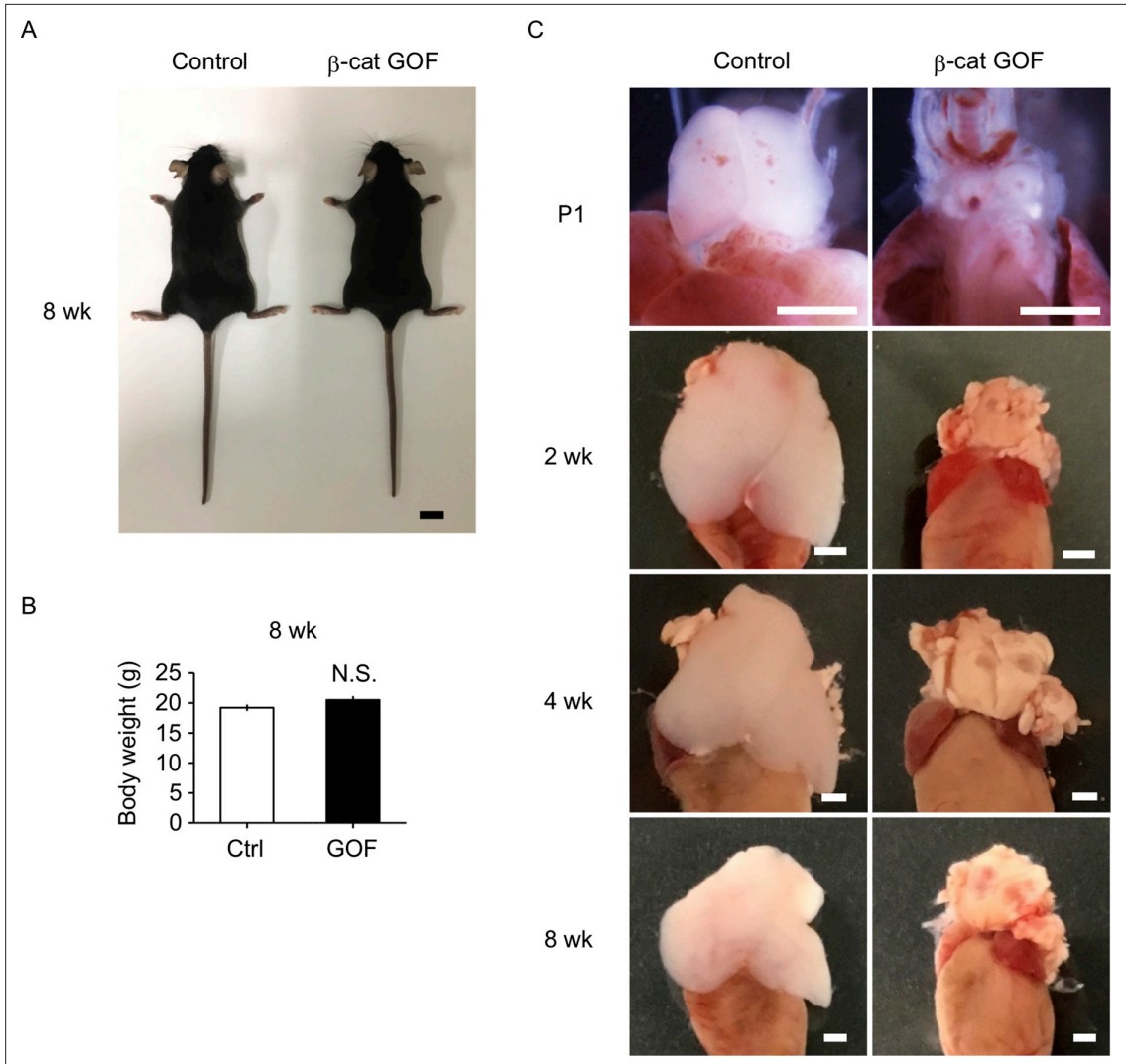

**Figure 3.** Thymic dysplasia in postnatal β-cat gain-of-function (GOF) mice. (**A**) Appearance of control mice and β-cat GOF mice at 8 wk. Bar: 1 cm. (**B**) Body weight of control mice and β-cat GOF mice at 8 wk (means and standard error of the means [SEMs], *n* = 4–6). N.S., not significant. (**C**) Appearance of the thymus in control mice and β-cat GOF mice at postnatal stages (P1–8 wk). Bar: 1 mm.

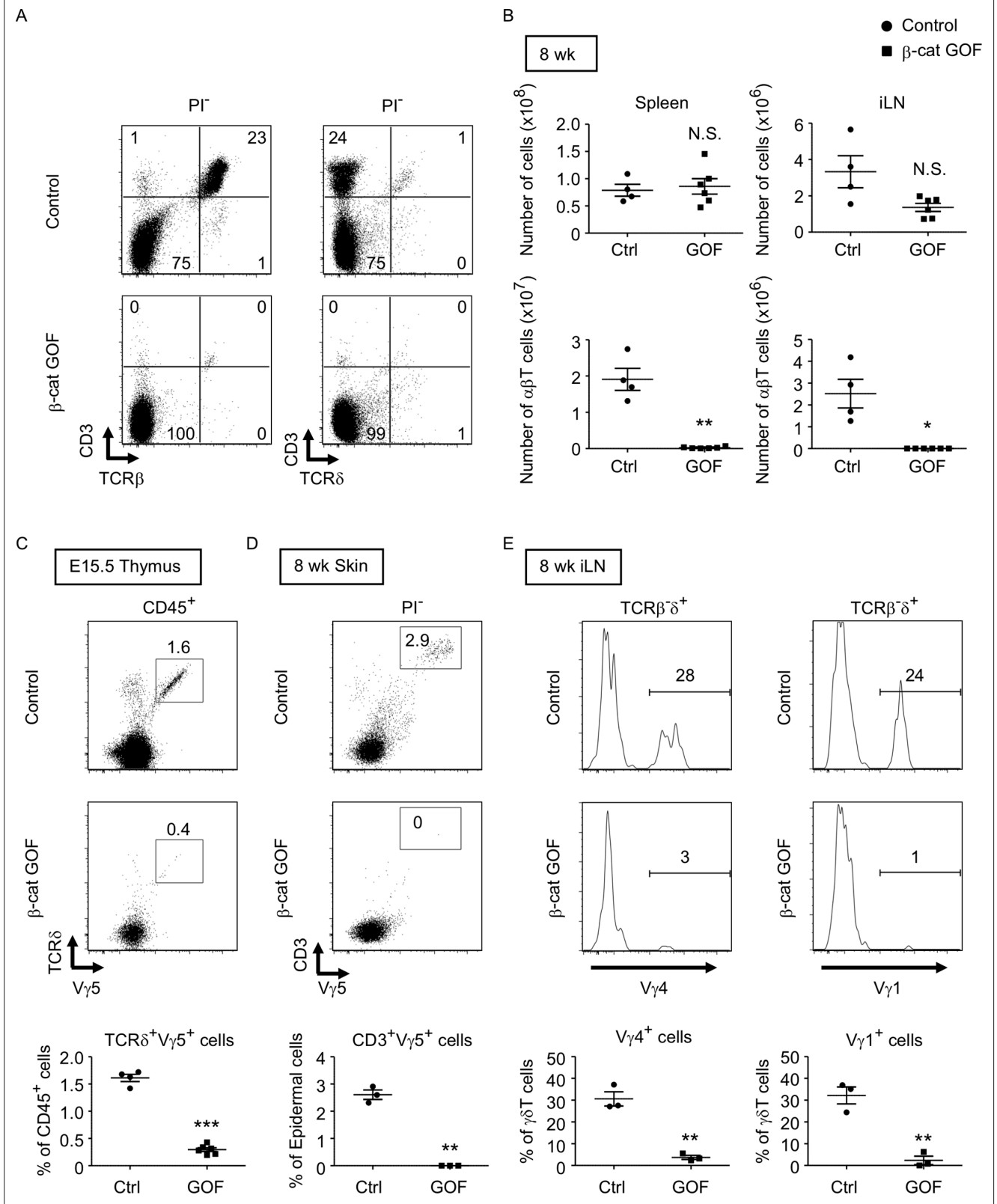

**Figure 4.** Loss of αβT and γδT cells in secondary lymphoid organs by the overexpression of β-catenin in thymic epithelial cells (TECs). (**A**) Flow cytometric analysis of splenocytes from control mice and β-cat gain-of-function (GOF) mice at 11 wk. Shown are representative dot plot profiles of CD3 and TCRβ expression (left) and CD3 and TCRδ expression (right) in PI⁻ viable cells. The numbers in dot plots indicate the frequency of cells within indicated area. (**B**) Cell numbers (means and standard error of the means [SEMs], n = 4–6) of indicated subpopulations in the spleen (left) and the iLN

*Figure 4 continued on next page*

**Figure 4 continued**

(right) from control and β-cat GOF mice at 8 wk are plotted. (**C**) Shown are representative dot plots of TCRδ and Vγ5 expression in CD45[+] cells in the thymus from control mice and β-cat GOF mice at E15.5. The frequency of TCRδ[+]Vγ5[+] cells is plotted (mean and SEMs; n = 4–6). (**D**) Flow cytometric analysis of dendritic epidermal T cells (DETCs) in the skin epidermis from control mice and β-cat GOF mice at 8 wk. Shown are representative dot plot profiles of CD3 and Vγ5 expression in epidermal cells. The numbers in dot plots indicate the frequency of cells within indicated area. The frequency of CD3[+]Vγ5[+] cells in epidermal cells is plotted (means and SEMs; n = 3). (**E**) Histograms for Vγ4 and Vγ1 expression in TCRβ[-]δ[+] cells in the iLN from control mice and β-cat GOF mice at 8 wk are shown. The frequency of Vγ4[+] and Vγ1[+] cells is plotted (means and SEMs; n = 3). *p < 0.05; **p < 0.01; ***p < 0.001; N.S., not significant.

(**Figure 5D, E**), indicating that β5t-Cre-mediated loss of β-catenin in TECs did not alter embryonic thymus development by E15.5.

## β-Catenin in TECs is required for optimal size of postnatal thymus

β-Cat LOF mice grew to adulthood with normal hair coat (**Figure 6A**). However, thymus size in β-cat LOF mice was noticeably reduced in comparison to that in control mice during the postnatal period (**Figure 6B**), with concomitant reductions in thymus weight but not body weight (**Figure 6C**). Flow cytometric analysis revealed that the β-catenin expression was effectively reduced by 96% in UEA1[-]Ly51[+] cTECs and by 94% in UEA1[+]Ly51[-] mTECs in β-cat LOF mice at 2 wk old (**Figure 7A**). Accordingly, quantitative RT-PCR analysis showed marked reduction of β-catenin mRNA (*Ctnnb1*) expression in cTECs and mTECs of β-cat LOF mice (**Figure 7B**, **Figure 7—figure supplement 1**).

In the thymus of β-cat LOF mice, the frequency and the number of cTECs were decreased compared with control mice (**Figure 7C, D**). *Axin2* expression in cTECs was significantly reduced in β-cat LOF mice, indicating that Wnt/β-catenin signals were efficiently impaired in cTECs from β-cat LOF mice. However, the expression of *Foxn1*, *Il7*, and *Cxcl12*, which encode functionally important molecules in cTECs, was not altered in cTECs from β-cat LOF mice (**Figure 7B**). On the other hand, despite that the expression of β-catenin in mTECs was markedly reduced in β-cat LOF mice, the cellularity of mTECs was not altered in β-cat LOF mice (**Figure 7C, D**), and the expression of *Axin2* was not reduced in mTECs from β-cat LOF mice (**Figure 7B**). The expression of functionally relevant molecules in mTECs, such as *Aire*, *Tnfrsf11a*, and *Ccl21a*, was comparable between β-cat LOF and control mTECs (**Figure 7B**). In agreement with the results of quantitative RT-PCR analysis, there were no apparent abnormalities in the corticomedullary compartmentalization of the thymus, as evidenced by the detection of β5t[+] cTECs in the cortex and CCL21[+] mTECs and AIRE[+] mTECs in the medulla, in β-cat LOF mice (**Figure 7E**). These results indicate that the β-catenin in TECs is required for optimizing postnatal thymus size, mainly affecting the number of cTECs, but is dispensable for the development of the normal number of mTECs.

To gain an insight into the mechanism underlying the reduced number of cTECs but not mTECs, we examined transcriptome profiles by RNA sequencing analysis using cTECs and mTECs isolated from the thymus of control and β-cat LOF mice at 2 wk. Using the method for RNA sequencing analysis of cTECs and mTECs (**Ohigashi et al., 2019**), we detected 92 genes that were significantly altered in β-cat LOF cTECs with the false discovery rate adjusted p value of less than 0.05; there were 11 downregulated genes and 81 upregulated genes compared with control cTECs. We also detected 91 genes that were significantly altered in β-cat LOF mTECs; there were 53 downregulated genes and 38 upregulated genes compared with control mTECs (**Figure 8A**). Thus, only less than 100 genes were significantly altered in cTECs and mTECs due to LOF of β-catenin, in agreement with the finding of no remarkable alteration in the expression of many functionally relevant molecules in cTECs and mTECs by quantitative RT-PCR analysis (**Figure 7B**). Nonetheless, we found that a vast majority of the genes that were significantly affected by LOF of β-catenin were different between cTECs and mTECs except *Adh1* and *Pglyrp1* (**Figure 8B**). *Adh1* was upregulated in β-cat LOF cTECs and downregulated in β-cat LOF mTECs, whereas *Pglyrp1* was upregulated by β-cat LOF in both cTECs and mTECs (**Figure 8B**). Notably, *Cdkn1a* was elevated specifically in cTECs but not in mTECs in β-cat LOF mice (**Figure 8A**). The expression of *Cdkn1a*, which encodes cyclin-dependent kinase (CDK) inhibitor p21 (**Abbas and Dutta, 2009**), is linked to β-catenin activity (**van de Wetering et al., 2002**; **Chee et al., 2021**; **Dagg et al., 2021**). The upregulation of *Cdkn1a* in β-cat LOF cTECs compared with control cTECs was confirmed by quantitative RT-PCR analysis (**Figure 8C**). We noticed no remarkable difference in other CDK family genes, including *Cnnb1*, *Cnnb2*, and *Cnnd1*, in cTECs and mTECs from β-cat LOF mice

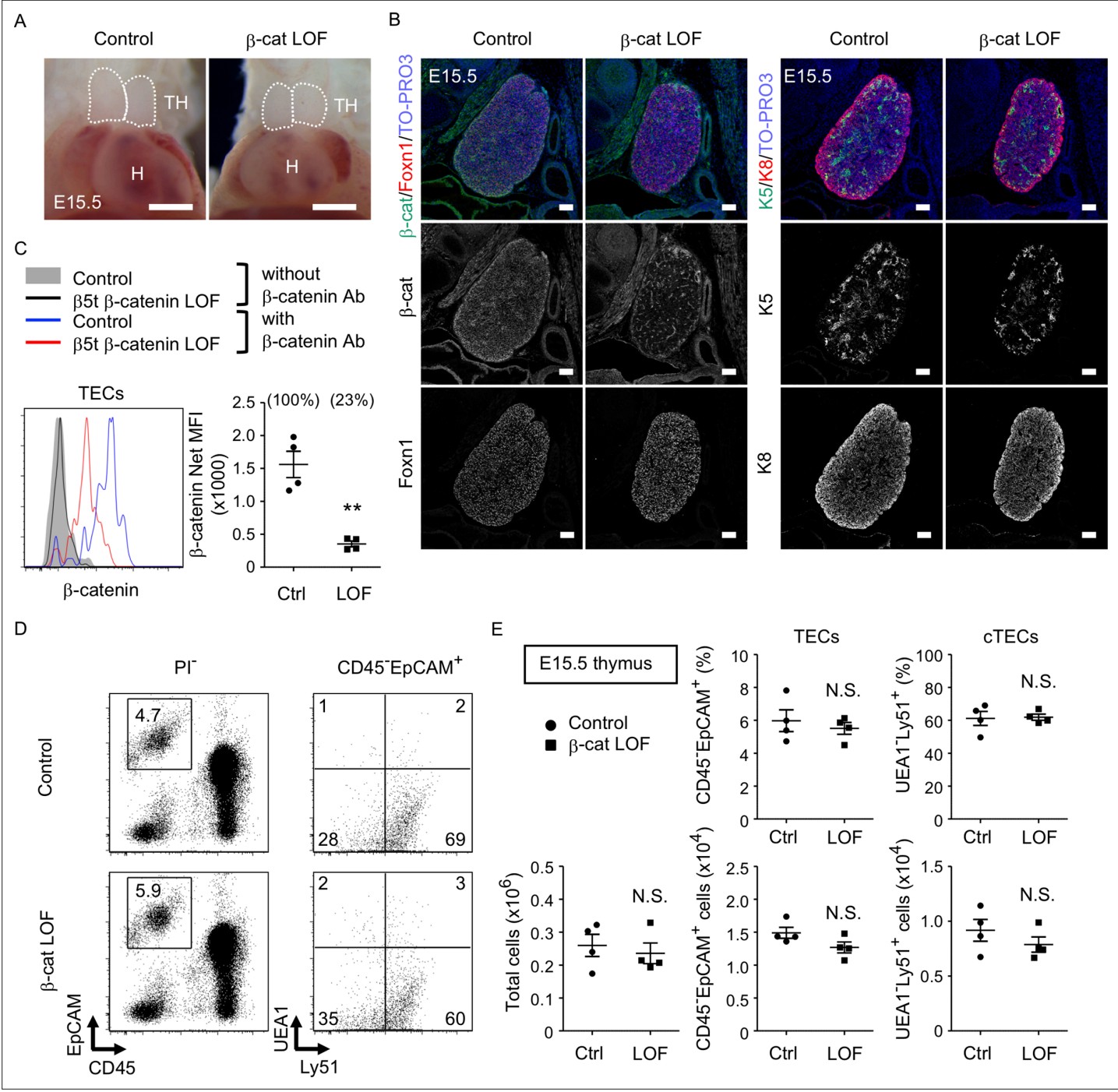

**Figure 5.** No apparent change in thymus development in β-cat loss-of-function (LOF) embryos. (**A**) Thoracic cavity of control mice and β-cat LOF mice at E15.5. Dotted lines show the outline of the thymic primordium. Representative data from three independent experiments are shown. TH: thymus, H: heart. Bar: 1 mm. (**B**) Immunofluorescence staining for β-catenin and Foxn1 (left) or K5 and K8 (right) on sagittal sections of the thymus from control mice and β-cat LOF mice at E15.5. Shown are merged images with nuclear counterstaining (TO-PRO3) (top) and images obtained in each channel (middle, bottom). Representative data from three independent experiments are shown. Bar: 100 μm. (**C**) Intracellular staining of β-catenin in CD45⁻EpCAM⁺ thymic epithelial cells (TECs) from control mice and β-cat LOF mice at E15.5. Histograms show β-catenin expression in control TECs (blue line) and β-cat LOF TECs (red line). Shaded area and black line represent the fluorescence in the absence of anti-β-catenin antibody in control TECs and β-cat LOF TECs, respectively. Plots show net median fluorescence intensity (MFI) values for β-catenin (means and standard error of the means [SEMs], *n* = 4). The numbers in parentheses indicate percentage of control value. (**D**) Flow cytometric analysis of enzyme-digested thymic cells from indicated mice at E15.5. Shown are profiles of EpCAM and CD45 expression in PI⁻ viable cells (left) and UEA1 reactivity and Ly51 expression in CD45⁻EpCAM⁺ cells (right). The numbers in dot plots indicate the frequency of cells within indicated area. (**E**) Plots show the number of total thymic cells (left) and the frequency and the

*Figure 5 continued on next page*

*Figure 5 continued*

number of total TECs (middle) and cortical thymic epithelial cell (cTECs; right) from control mice and β-cat LOF mice at E15.5 (means and SEMs, *n* = 4). **p < 0.01; N.S., not significant.

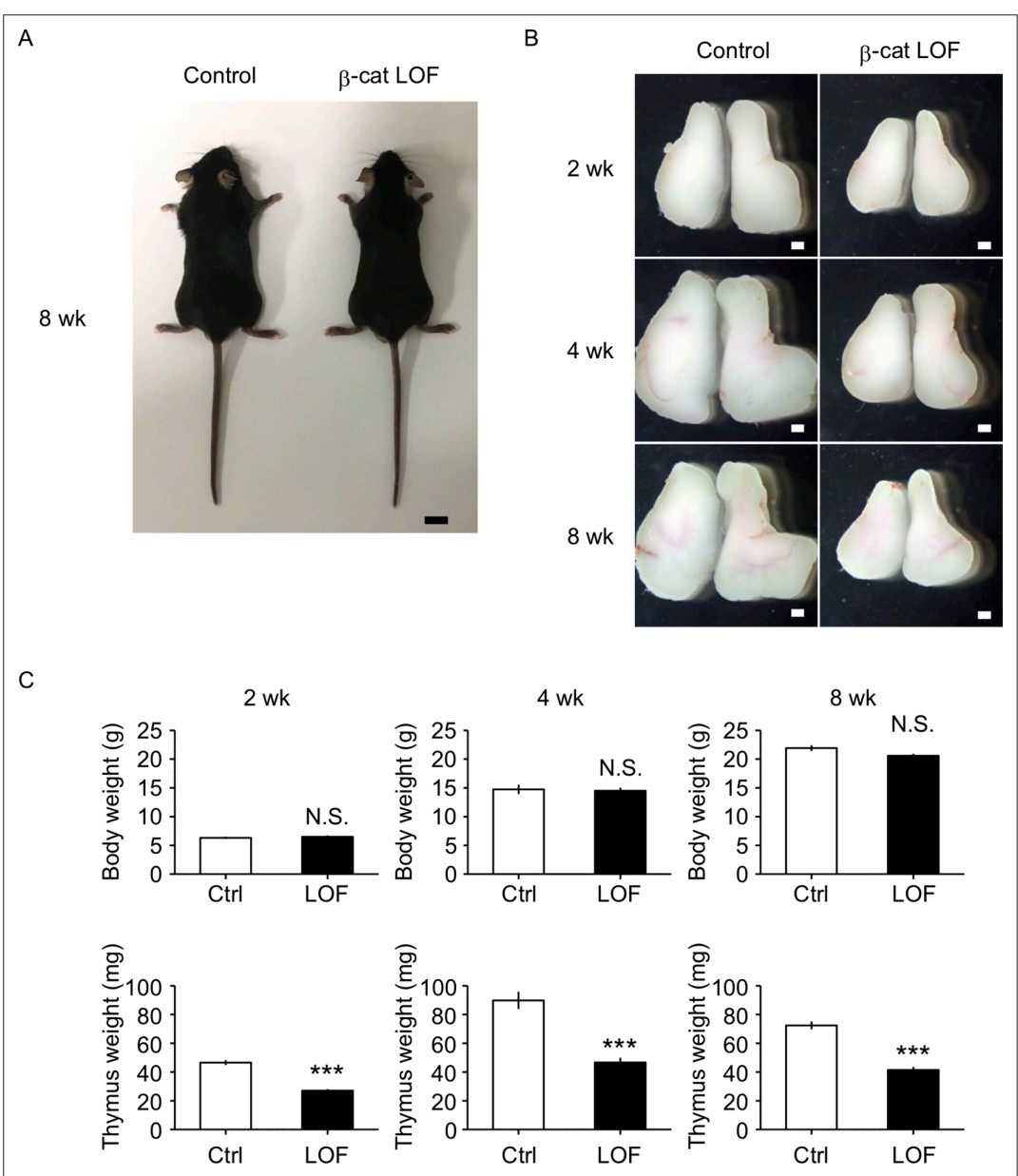

**Figure 6.** The reduction of thymus size in postnatal β-cat loss-of-function (LOF) mice. (**A**) Appearance of control mice and β-cat LOF mice at 8 wk. Bar: 1 cm. (**B**) Appearance of the thymus from control mice and β-cat LOF mice at postnatal stages (2–8 wk). Representative data from at least three independent experiments are shown. Bar: 1 mm. (**C**) Bars show body weight (top) and thymus weight (bottom) at 2 wk (left, *n* = 3), 4 wk (middle, *n* = 4), and 8 wk (right, *n* = 5) in control mice and β-cat LOF mice (mean and standard error of the means [SEMs]). ***p < 0.001; N.S., not significant.

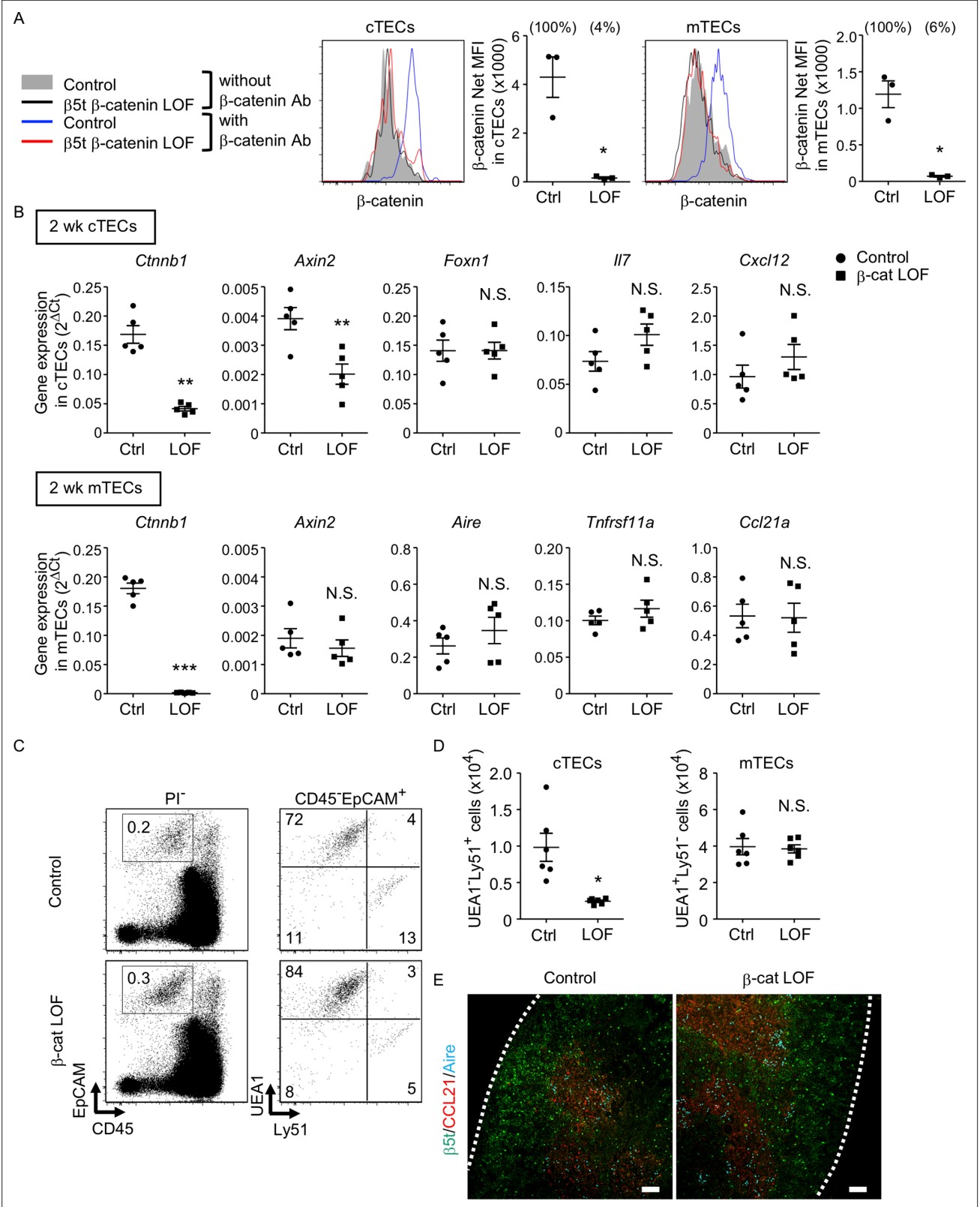

**Figure 7.** Characteristics of thymic epithelial cells (TECs) in postnatal β-cat loss-of-function (LOF) mice. (**A**) Intracellular staining of β-catenin in UEA1⁻Ly51⁺ cortical thymic epithelial cells (cTECs; left) and UEA1⁺Ly51⁻ medullary thymic epithelial cells (mTECs; right) from control mice and β-cat LOF mice at 2 wk. Histograms show β-catenin expression in cTECs and mTECs from control mice (blue line) and β-cat LOF mice (red line). Shaded area and black line represent the fluorescence in the absence of anti-β-catenin antibody in control TECs and β-cat LOF TECs, respectively. Plots show net

*Figure 7 continued*

MFI values for β-catenin in cTECs and mTECs (means and standard error of the means [SEMs], *n* = 3). The numbers in parentheses indicate percentage of control value. (**B**) Quantitative RT-PCR analysis of mRNA expression levels (means and SEMs, *n* = 5) of indicated genes relative to *Gapdh* levels in UEA1⁻Ly51⁺ cTECs (top) and UEA1⁺Ly51⁻ mTECs (bottom) in the thymus of control mice and β-cat LOF mice at 2 wk. (**C**) Flow cytometric analysis of enzyme-digested thymic cells from control mice and β-cat LOF mice at 2 wk. Shown are representative profiles of EpCAM and CD45 expression in PI⁻ viable cells (left) and UEA1 reactivity and Ly51 expression in CD45⁻EpCAM⁺ viable cells (right). The numbers in dot plots indicate the frequency of cells within indicated area. (**D**) Plots show the number (means and SEMs, *n* = 6) of cTECs and mTECs in the thymus from control mice and β-cat LOF mice at 2 wk. (**E**) Immunofluorescence analysis of β5t (green), CCL21 (red), and Aire (cyan) on transverse sections of thymus from control mice and β-cat LOF mice at 2 wk. Representative data from three independent experiments are shown. Bar: 100 μm. Ctrl: Control, LOF: β-cat LOF. *p < 0.05; **p < 0.01; ***p < 0.001; N.S., not significant.

The online version of this article includes the following figure supplement(s) for figure 7:

**Figure supplement 1.** The purity of isolated cortical thymic epithelial cells (cTECs) and medullary thymic epithelial cells (mTECs) for quantitative RT-PCR analysis.

(data not shown). These results suggest that the upregulation of *Cdkn1a* may contribute to the selective reduction in the number of cTECs by LOF of β-catenin.

## β-Catenin in TECs affects postnatal thymocyte number

We next examined thymocytes in postnatal β-cat LOF mice. The number of thymocytes was reduced in β-cat LOF mice compared with control at 2, 4, and 8 wk of age (*Figure 9A*). The frequency of DN, DP, CD4⁺CD8⁻, and CD4⁻CD8⁺ thymocytes, as well as the frequency of TCRβ⁻δ⁺ thymocytes, was not altered in β-cat LOF mice at 8 wk (*Figure 9B*). As a result of the reduction in the number of total thymocytes, the number of thymocyte subsets defined by CD4 and CD8, as well as the number of TCRβ⁻δ⁺ thymocytes, was reduced in β-cat LOF mice, although the reduction in the number of CD4⁺CD8⁻ thymocytes in these mice was not statistically significant (p = 0.069) (*Figure 9C*). We also noted that the frequency of DN subsets defined by CD44 and CD25 was unchanged, although the number of all four DN subsets was significantly reduced in β-cat LOF mice (*Figure 9D*). On the other hand, in contrast to the significant reduction in thymocyte numbers, the number of αβT cells and γδ T cells was not significantly altered in the secondary lymphoid organs, including the spleen and the iLN (*Figure 9E, F*, data not shown).

Our results indicate that the β-catenin in TECs is not essential for the generation of functional TECs that support T-cell development. However, the loss of β-catenin in TECs results in the reduction in the number of cTECs, which leads to the reduction in the number of thymocytes during the postnatal period.

## Age-associated thymic involution occurs in LOF of β-catenin in TECs

We next examined the phenotype of the thymus in β-cat LOF mice at 6 months old (6 mo), in which the involution of the thymus was apparent. The weight of the thymus decreased in control mice at 6 mo in comparison with those at 4 weeks old (4 wo) and 8 wo (*Figures 6C and 10A*). Similarly, the thymus weight decreased in β-cat LOF mice at 6 mo compared with that at 4 wo (*Figures 6C and 10A*). Accordingly, the thymus at 6 mo was still smaller in β-cat LOF mice than control mice (*Figure 10A, B*). The number of thymocytes was reduced upon the thymic involution in β-cat LOF mice (*Figures 9A and 10C*), although the difference in thymocyte number became insignificant between control mice and β-cat LOF mice at 6 mo (*Figure 10C*). The number of cTECs in β-cat LOF mice at 6 wo remained smaller than the control number, whereas the number of mTECs was equivalent to that of control at 6 mo (*Figure 10 D, E*). There were no apparent differences in the corticomedullary architecture of the thymus between control mice and β-cat LOF mice at 6 mo (*Figure 10F*). These results indicate that age-associated thymic involution is detectable in β-cat LOF mice.

## LOF of β-catenin in TECs delays but does not abrogate recovery of thymus from stress-induced involution

Because the abnormality in the thymus of β-cat LOF mice appeared much less severe than that of β-cat GOF mice, we examined how stress-induced injury would affect the thymus in β-cat LOF mice. To do so, β-cat LOF mice were treated with polyinosinic:polycytidylic acid (poly(I:C)), a synthetic analog of double-stranded RNA, which caused injury in TECs (*Demoulins et al., 2008*; *Papadopoulou et al.,*

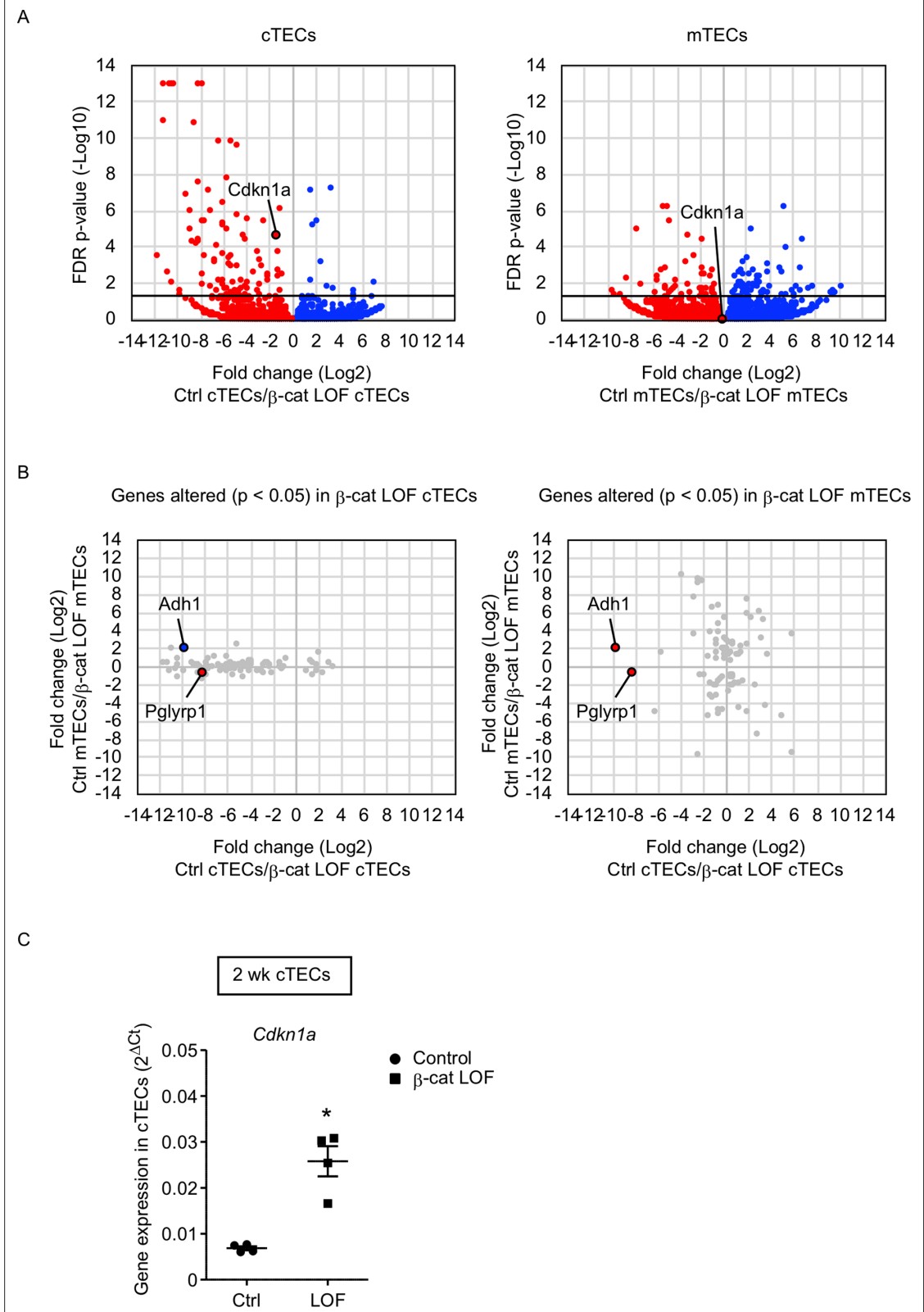

**Figure 8.** RNA sequencing analysis of cortical thymic epithelial cells (cTECs) and medullary thymic epithelial cells (mTECs) isolated from β-cat loss-of-function (LOF) mice. (**A**) Volcano plot analysis of mRNA expression for cTECs (left) and mTECs (right) isolated from control (Ctrl) mice and β-cat LOF mice at 2 wk. Detected genes are plotted as log2 fold change (Ctrl/β-cat LOF) versus −log10 false discovery rate (FDR) p value. Bold horizontal lines in the plot show the p value of 0.05. (**B**) Correlation plot analysis for the genes differently altered (p < 0.05) in β-cat LOF cTECs (left) and β-cat LOF mTECs

*Figure 8 continued on next page*

*Figure 8 continued*

(right). Log2 fold changes of genes altered between Ctrl cTECs and β-cat LOF cTECs are plotted against log2 fold changes of genes altered between Ctrl mTECs and β-cat LOF mTECs. (**C**) Quantitative RT-PCR analysis of *Cdkn1a* mRNA expression normalized to *Gapdh* levels in UEA1⁻Ly51⁺ cTECs isolated from the thymus of control mice and β-cat LOF mice at 2 wk (means and standard error of the means [SEMs], *n* = 4). *p < 0.05.

*2011*, *Figure 11A*). We found that both control and β-cat LOF mice showed transient thymic involution that was evident from the loss of thymus weight and total thymic cell number 4 days after the first poly(I:C) administration (*Figure 11B, C*). Subsequently, however, the involuted thymus recovered to its pretreatment size by days 18 and 25 in both control and β-cat LOF mice (*Figure 11B, C*), indicating that the small thymus in postnatal β-cat LOF mice retained the capability to recover from the poly(I:C)-mediated thymic injury. Indeed, the cTECs and mTECs recovered to their numbers before the treatment (*Figure 11D*). Interestingly, we noticed a delay in the recovery of the thymus in β-cat LOF mice compared with that in control mice (*Figure 11B, C*). The weight and the cell number of the thymus were still significantly lower on day 11 than day 0 in β-cat LOF mice, unlike the recovery to the pretreatment levels on day 11 in control mice (*Figure 11B, C*). Accordingly, the number of DP thymocytes remained slightly but significantly reduced even on day 25 in β-cat LOF mice (*Figure 11E*). These results indicate that LOF of β-catenin in TECs delays the recovery of the thymus from the poly(I:C)-induced thymic involution, but does not fully abolish the capability of the thymus to recover from the poly(I:C)-induced thymic injury.

## Discussion

The important role played by Wnt/β-catenin signaling in thymus organogenesis has been well appreciated. However, the role of Wnt/β-catenin signaling in TECs with regard to postnatal immune system development remains unclear due to the difficulty of performing an in vivo analysis without generating extrathymic side effects. Most typically, the conventional 'TEC-specific' Foxn1-Cre-mediated genetic manipulation of Wnt/β-catenin signaling molecules causes neonatal lethality of the animals due to side effects in the skin epidermis. In the present study, we employed the recently devised β5t-Cre, which enables highly efficient and specific genetic manipulation in TECs without generating side effects in the skin epidermis or other cells in the body. By analyzing β5t-Cre-mediated deficiency in β-catenin highly specific in TECs, we found that the TEC-specific loss of β-catenin causes postnatal reduction in the number of cTECs, which leads to the reduction in the number of thymocytes. Nonetheless, the TEC-specific loss of β-catenin did not cause an arrest in the development of the thymus, including the corticomedullary architecture, and the subsequent generation of T cells, indicating that the β-catenin in TECs is dispensable for the development of cTECs and mTECs as well as of postnatal T cells. In contrast, our results also showed that β5t-Cre-mediated constitutive activation of β-catenin in TECs causes dysplasia in thymus organogenesis and loss of thymic T-cell development. These results suggest that fine-tuning of β-catenin-mediated signaling activity in TECs is required for optimal development and maintenance of functional thymus. Thus, β-catenin-mediated signaling in TECs needs to be maintained at an intermediate level to prevent thymic dysplasia induced by its excessive signals, whereas no or extremely low β-catenin signals cause hypoplasia of the postnatal thymus. The need for fine-tuning of β-catenin during the development of the central nervous system and other organs, such as bone, heart, intestine, liver, lung, mammary gland, pancreas, reproductive organs, skin, and tooth, was reported (*Grigoryan et al., 2008*). Our LOF and GOF experiments in TECs established that the thymus is included in tissues that require fine-tuning of β-catenin during the development, and revealed that the fine-tuning of β-catenin in TECs is required for supporting postnatal T-cell development in the thymus.

Previous conditional gene targeting in TECs was performed using Cre-deleter mouse lines with any one of the promoter sequences for Foxn1, K5, and K14. However, extrathymic side effects were induced in epithelial cells other than TECs in the Cre-deleter mouse lines. For example, Foxn1-Cre-mediated genetic alteration was detectable in the skin in addition to TECs (*Zuklys et al., 2009*), and mice rendered β-catenin deficient using Foxn1-Cre died within a few days of birth due to defects in skin development (*Swann et al., 2017*). In contrast, the β5t-locus knockin Cre-deleter mice used in the present study enabled highly efficient and specific genetic manipulation in TECs without any side effects in other organs including the skin. β5t is abundant in cTECs but not detectable in other cells

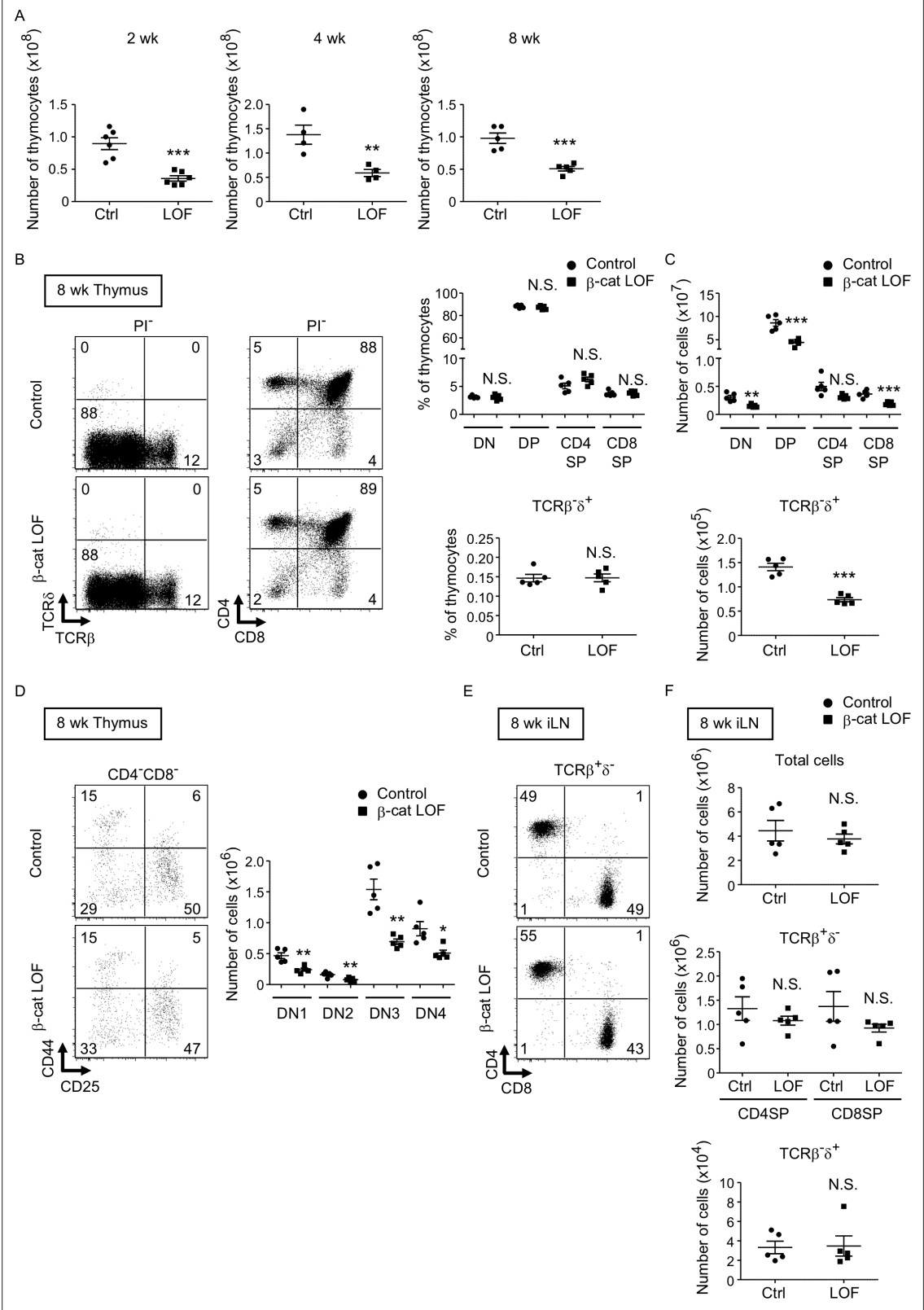

**Figure 9.** Reduced total thymocyte production in β-cat loss-of-function (LOF) mice during the postnatal development. (**A**) Plots show the number (means and standard error of the means [SEMs], n = 4–6) of total thymocytes in the thymus of indicated mice at 2 wk (left), 4 wk (middle), and 8 wk (right). (**B**) Flow cytometric analysis of thymocytes from control mice and β-cat LOF mice at 8 wk. Shown are representative dot plot profiles of TCRβ and TCRδ expression and CD4 and CD8 expression in PI⁻ viable cells. The numbers in dot plots indicate the frequency of cells within indicated area. Plots show

*Figure 9 continued on next page*

*Figure 9 continued*

the frequency of indicated thymocyte subpopulations (means and SEMs, *n* = 5). (**C**) Cell numbers (means and SEMs, *n* = 5) of indicated subpopulations in the thymus from control mice and β-cat LOF mice at 8 wk. (**D**) The frequency and number of DN thymocyte subsets in indicated mice at 8 wk. Dot plots (left) show representative CD44 and CD25 profiles in CD4⁻CD8⁻ viable thymocytes. The numbers in dot plots indicate the frequency of cells within indicated area. Right panel shows cell numbers (means and SEMs, *n* = 5) of indicated DN subsets in the thymus from control mice and β-cat LOF mice. (**E**) Flow cytometric analysis of lymphocytes in the iLN from control mice and β-cat LOF mice at 8 wk. Shown are representative dot plot profiles of CD4 and CD8 expression in TCRβ⁺δ⁻ viable cells. The numbers in dot plots indicate the frequency of cells within indicated area. (**F**) Cell numbers (means and SEMs, *n* = 4–5) of indicated subpopulations in the iLN from control mice and β-cat LOF mice at 8 wk. *p < 0.05; **p < 0.01; ***p < 0.001; N.S., not significant.

including mTECs (*Ripen et al., 2011*). However, all mTECs are derived from bipotent β5t-expressing TEC progenitors (*Ohigashi et al., 2013*). Therefore, β5t-iCre enables efficient and conditional genetic manipulation of both cTECs and mTECs (*Ohigashi et al., 2015*). A conceptually similar gene targeting strategy has been widely employed to study two major T-cell lineages of CD4 and CD8 T cells, as CD4-Cre is useful to delete floxed sequences in both CD4 and CD8 T cells (*Sharma and Zhu, 2014*). In agreement with a recent study using the same β5t-iCre (*Barthlott et al., 2021*), our study demonstrated the highly efficient deletion of *Ctnnb1* floxed sequence in both cTECs and mTECs (*Figure 7A*).

Our β-cat LOF and GOF mice using the TEC-specific β5t-iCre grew postnatally without perinatal lethality or skin defects. Interesting, both of our β-cat LOF and GOF mice exhibited thymic phenotypes different from previously observed phenotypes (*Figure 12*). The β5t-Cre-mediated GOF of β-catenin caused dysfunction of the embryonic thymus, in line with previous β-cat GOF studies using Foxn1-Cre (*Zuklys et al., 2009*; *Swann et al., 2017*). However, unlike those previous studies, defective migration of the thymus into the thoracic cavity during the embryogenesis was not observed in our β5t-Cre-mediated β-cat GOF mice. Therefore, our study clarified that the β-cat GOF-mediated thymic dysfunction is not due to impaired migration of the thymic primordium from the pharyngeal region to the chest cavity. Instead, the difference in migration phenotype may be due to the different onset of Cre activity in TECs between different Cre lines. The Foxn1-mediated direct transcriptional regulation of β5t and a slightly later expression of β5t than Foxn1 (*Ripen et al., 2011*; *Uddin et al., 2017*) may result in Foxn1-Cre-mediated, but not β5t-Cre-mediated, β-cat GOF interference with thymic migration toward the thoracic cavity.

More prominently, unlike Foxn1-Cre-mediated β-cat LOF mice, β5t-Cre-mediated β-cat LOF mice did not die perinatally but grew to adulthood, which enabled the analysis of postnatal T-cell development in TEC-specific β-catenin-deficient thymus. Our results demonstrated that the loss of β-catenin in TECs impacts neither thymic architecture nor the expression of functionally relevant molecules in cTECs and mTECs. Rather, we found that the β5t-Cre-mediated LOF of β-catenin reduces the number of postnatal cTECs. We further found that the number of postnatal thymocytes in the β-cat LOF thymus is reduced by half compared with that in control thymus, although the T-cell development is not disturbed. Thus, our analysis of β5t-Cre-mediated β-cat LOF mice revealed that the β-catenin signaling in TECs is dispensable for TEC development and T-cell development, but is required for the provision of cTECs, which support postnatal T-cell generation.

Our results indicated that β5t-Cre-mediated β-catenin-deficient mice exhibit postnatal reduction in the number of cTECs but not mTECs and that their expression of *Axin2* is reduced only in cTECs but not in mTECs, implying the differential responsiveness to β-catenin signaling between cTECs and mTECs. The selective effect on cTECs was detectable in mice at 6 mo, in which age-associated thymic involution was apparent. Transcriptome profiling by RNA sequencing analysis confirmed the differential responsiveness between cTECs and mTECs to LOF of β-catenin, as represented by the cTEC-specific elevation of *Cdkn1a* in β-cat LOF mice. The increase in *Cdkn1a* specifically in cTECs may lead to the reduction in the number of cTECs rather than mTECs. It is further interesting to note that the reduced number of cTECs but not mTECs is associated with an equivalent reduction in the number of immature thymocytes in the thymic cortex and the number of mature thymocytes in the thymic medulla, suggesting that the number of mature thymocytes generated in the thymus is correlated with the number of cTECs rather than the number of mTECs.

Our results showed that GOF of β-catenin in TECs results in the decrease in the expression of TEC-specific molecules and the increase in the expression of terminally differentiated keratinocytes, in agreement with previous results using Foxn1-Cre-mediated β-cat GOF mice (*Zuklys et al., 2009*).

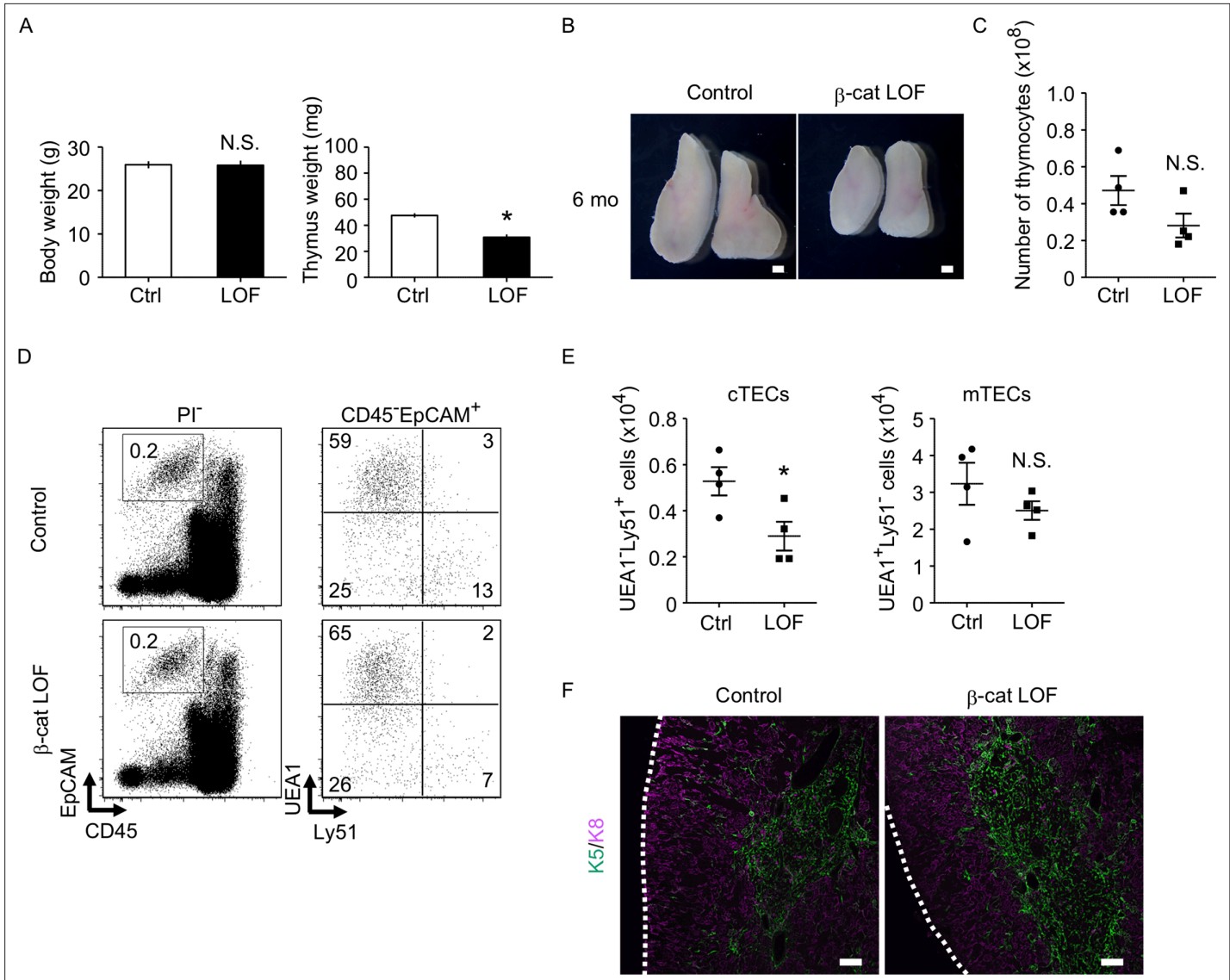

**Figure 10.** The thymus of β-cat loss-of-function (LOF) mice at 6 mo. (**A**) Bars show body weight (left) and thymus weight (right) at 6 mo in control female mice and β-cat LOF female mice (means and standard error of the means [SEMs], *n* = 4). (**B**) Appearance of thymus from control mice and β-cat LOF mice at 6 mo. Representative data from four independent experiments are shown. Bar: 1 mm. (**C**) Plots show the number (means and SEMs, *n* = 4) of total thymocytes in the thymus from control mice and β-cat LOF mice at 6 mo. (**D**) Flow cytometric analysis of enzyme-digested thymic cells from control mice and β-cat LOF mice at 6 mo. Shown are representative profiles of EpCAM and CD45 expression in PI⁻ viable cells (left) and UEA1 reactivity and Ly51 expression in CD45⁻EpCAM⁺ viable cells (right). The numbers in dot plots indicate the frequency of cells within indicated area. (**E**) Plots show the number (means and SEMs, *n* = 4) of cortical thymic epithelial cells (cTECs) and medullary thymic epithelial cells (mTECs) in the thymus from control mice and β-cat LOF mice at 6 mo. (**F**) Immunofluorescence analysis of K5 (green) and K8 (magenta) on transverse sections of thymus from control mice and β-cat LOF mice at 6 mo. Representative data from three independent experiments are shown. Bar: 100 μm. Ctrl: Control, LOF: β-cat LOF. *p < 0.05; N.S., not significant.

It is possible that TECs retain the potential to transdifferentiate into epidermal keratinocytes in the presence of an excessive amount of β-catenin, as a similar transdifferentiation of epithelial cells upon β-catenin overexpression was reported in other tissues, such as mammary gland and prostate (*Miyoshi et al., 2002*; *Bierie et al., 2003*). However, our data suggest a slightly different scenario because keratin 1 (*Krt1*) and keratin 10 (*Krt10*) genes, which are expressed during the cornification of terminally differentiated keratinocytes (*Candi et al., 2005*), were not detectable in β5t-Cre-mediated β-cat LOF TECs at E15.5 (data not shown). We suspect that β-cat GOF TECs are not exactly equivalent to the terminally differentiated keratinocytes, although they partially share gene expression profiles.

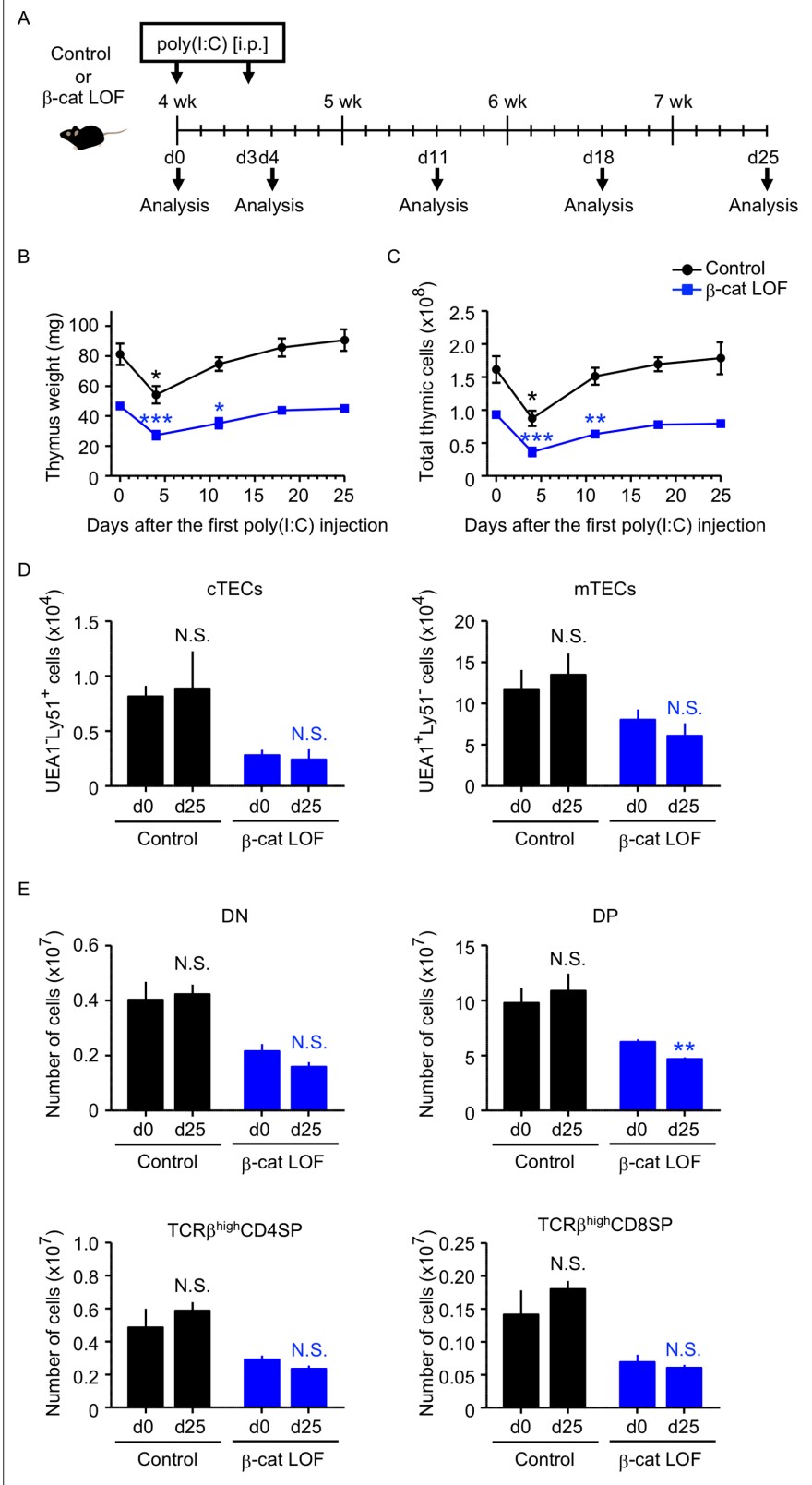

**Figure 11.** Thymus responses to poly(I:C) administration in β-cat loss-of-function (LOF) mice. (**A**) Schematic diagram of poly(I:C) administration. Control mice or β-cat LOF mice at 4 wk were intraperitoneally injected with 250 µg of poly(I:C) on days 0 and 3, and the thymus was analyzed on the indicated days after the first poly(I:C) injection. d, day. Time course of thymus weight (**B**) and total thymic cellularity (**C**) in control mice (black line) and

*Figure 11 continued on next page*

*Figure 11 continued*

β-cat LOF mice (blue line) after poly(I:C) injection (means and standard error of the means [SEMs], *n* = 3–5). (**D**) Bars show the number of cortical thymic epithelial cells (cTECs) and medullary thymic epithelial cells (mTECs) on day 0 (d0) and day 25 (d25) after poly(I:C) injection (means and SEMs, *n* = 4–5) in control mice (black) and β-cat LOF mice (blue). (**E**) Bars show the number of indicated thymocytes on d0 and d25 after poly(I:C) injection (means and SEMs, *n* = 4–5) in control mice (black) and β-cat LOF mice (blue). *p < 0.05; **p < 0.01; ***p < 0.001; N.S., not significant.

It is noted that the terminally differentiated mTEC subpopulations, including post-Aire mTECs and Hassall's corpuscles, express involucrin (*Yano et al., 2008*; *Nishikawa et al., 2010*; *White et al., 2010*) and loricrin (*Michel et al., 2017*). Therefore, it is further possible that the alteration of β-catenin-mediated signals in mature mTECs promotes the terminal differentiation of the mTECs or the deviation to give rise to Hassall's corpuscles.

Cellular interactions between TECs and neighboring cells, such as mesenchymal cells (*Shinohara and Honjo, 1997*; *Jenkinson et al., 2003*; *Jenkinson et al., 2007*; *Itoi et al., 2007*), endothelial cells (*Bryson et al., 2013*; *Wertheimer et al., 2018*), and thymocytes (*Holländer et al., 1995*; *Klug et al., 2002*), are important for thymus development. Wnts and Wnt signaling molecules are expressed in TECs and neighboring cells in various amounts (*Balciunaite et al., 2002*; *Pongracz et al., 2003*; *Osada et al., 2006*; *Heinonen et al., 2011*; *Ki et al., 2014*; *Brunk et al., 2015*; *Nitta et al., 2020*) and at various timings (*Osada et al., 2006*; *Kvell et al., 2010*; *Varecza et al., 2011*; *Griffith et al., 2012*; *Ki et al., 2014*). The presence of a variety of Wnts and Wnt signaling molecules in developing TECs may trigger the fine-tuning of β-catenin signaling activity to establish the development of diverse TEC subpopulations.

We think that the β-catenin-mediated regulation of TEC development is dependent on TCF/LEF-mediated canonical Wnt signaling but not on the adherent function of β-catenin, which plays a role

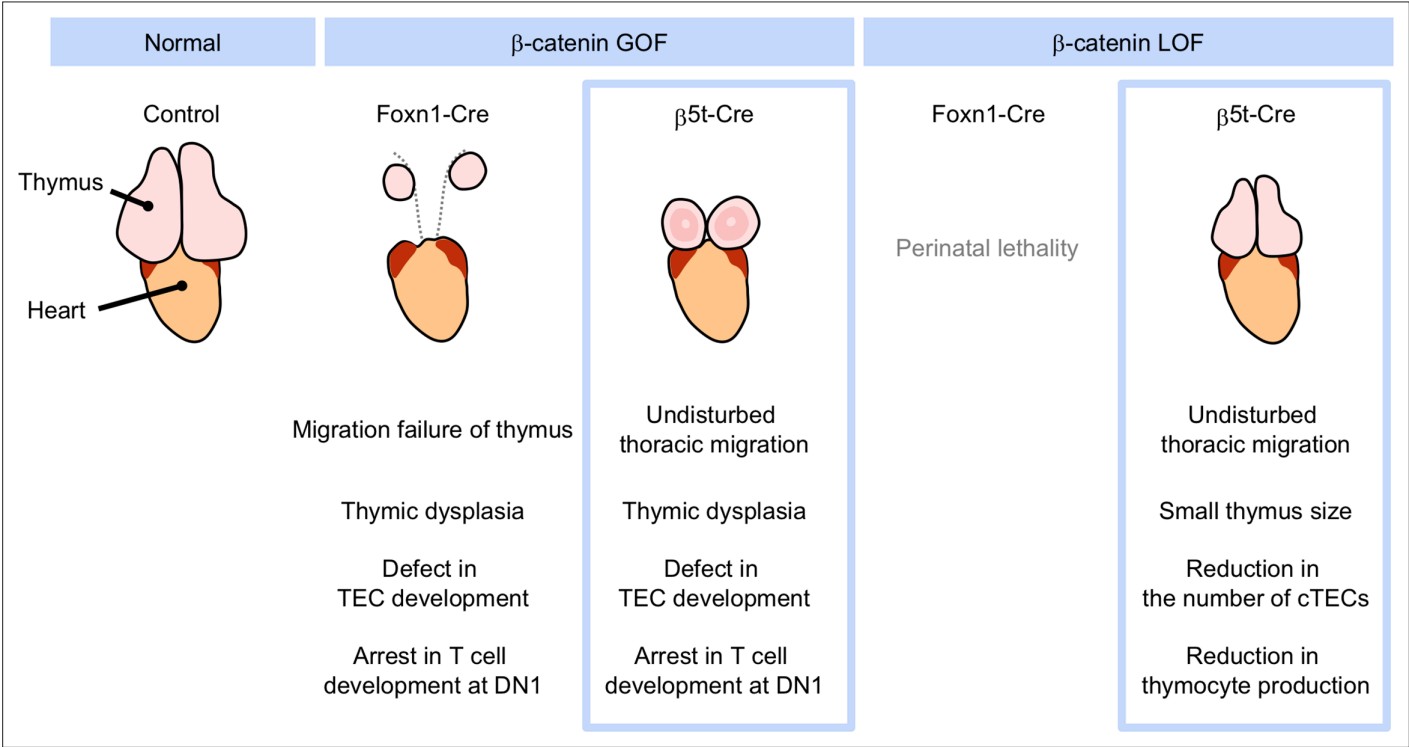

**Figure 12.** Phenotypic differences between conditional β-cat loss-of-function (LOF) and gain-of-function (GOF) mice during the postnatal period. GOF of β-catenin using Foxn1-Cre and β5t-iCre causes severe defect in thymic epithelial cell (TEC development and arrest in T-cell development at DN1), whereas Foxn1-Cre-mediated GOF of β-catenin only results in the failure of migration of thymic primordium into the thoracic cavity. LOF of β-catenin using Fonx1-Cre results in perinatal lethality, whereas LOF of β-catenin using β5t-iCre does not lead to reduced viability throughout development. β5t-Cre-mediated LOF of β-catenin results in reduction in thymus size and the number of cortical thymic epithelial cells (cTECs) and total thymocyte production during the postnatal period.

in the interaction between E-cadherin and α-catenin at adherens junctions (*Kemler, 1993*). It was previously shown that E-cadherin deficiency induced by Foxn1-Cre did not impact the number of TECs and was less effective than β-catenin deficiency in newborn mice (*Swann et al., 2017*). In addition, Foxn1-Cre-mediated E-cadherin-deficient adult mice exhibited no defect in the thymus (*Swann et al., 2017*), in contrast to β5t-Cre-mediated β-catenin-deficient mice, further suggesting that β-catenin-dependent signaling in TECs is likely independent of the adherent function of β-catenin.

The thymic microenvironment provided by TECs is essential for the development of functionally competent and self-tolerant T cells. Our in vivo genetic study of β-catenin specifically manipulated in TECs demonstrates that fine-tuning of β-catenin-mediated signaling in TECs is essential to maintain TEC function integrity for T-cell production during the embryogenesis and the postnatal period. Future studies of the function of Wnts and Wnt signaling components in TECs and neighboring cells should provide further insight into the mechanism for fine-tuning β-catenin signaling in TECs to support T-cell development in the thymus.

# Materials and methods

**Key resources table**

| Reagent type (species) or resource | Designation | Source or reference | Identifiers | Additional information |
|---|---|---|---|---|
| Strain, strain background (*Mus musculus*) | C57BL/6N | SLC Japan | RRID:MGI:5295404 | |
| Strain, strain background (*Mus musculus*) | β5t-iCre | *Ohigashi et al., 2013* | N/A | |
| Strain, strain background (*Mus musculus*) | *Ctnnb1* floxed (β-cat$^{fl}$) | *Brault et al., 2001* | RRID:IMSR_JAX:004152 | |
| Strain, strain background (*Mus musculus*) | *Ctnnb1* exon 3 floxed (β-cat$^{ex3fl}$) | *Harada et al., 1999* | N/A | |
| Strain, strain background (*Mus musculus*) | Rosa26-CAG-loxP-stop-loxP-tdTomato (R26R-tdTomato) | *Madisen et al., 2010* | RRID:IMSR_JAX:007914 | |
| Antibody | Anti-Foxn1 (Rabbit polyclonal) | *Itoi et al., 2006* | N/A | IHC (1:100) |
| Antibody | Anti-β5t (Rabbit polyclonal) | *Murata et al., 2007* | N/A | IHC (1:200) |
| Antibody | Anti-CCL21/6Ckine (Rat monoclonal, Clone 59106) | R&D systems | Cat# MAB457, RRID:AB_2259799 | IHC (1:10) |
| Antibody | Anti-Aire-eFluor 660 (Rat monoclonal, Clone 5H12) | e-Bioscience | Cat# 50-5934-82, RRID:AB_2574257 | IHC (1:20) |
| Antibody | Anti-β-catenin (Mouse monoclonal, Clone 14) | BD Transduction Laboratories | Cat# 610154, RRID:AB_397555 | IHC (1:200) FC (1:100) |
| Antibody | Anti-Keratin 5 (Chicken polyclonal, Clone Poly9059) | BioLegend | Cat# 905901, RRID:AB_2565054 | IHC (1:500) |
| Antibody | Anti-Keratin 8 (Mouse monoclonal, Clone IE8) | BioLegend | Cat# 904804, RRID:AB_2616821 | IHC (1:500) |
| Antibody | Anti-CD45-Biotin (Rat monoclonal, Clone 30-F11) | BioLegend | Cat# 103104, RRID:AB_312969 | IHC (1:100) |
| Antibody | Anti-mouse CD45-PE/Cy5 (Rat monoclonal, Clone 30-F11) | BioLegend | Cat# 103110, RRID:AB_312975 | IHC (1:100) |
| Antibody | Anti-CD45-eFluor 450 (Rat monoclonal, Clone 30-F11) | eBioscience | Cat# 48-0451-80, RRID:AB_1518807 | FC (1:200) |
| Antibody | CD45 MicroBeads (Rat monoclonal) | Miltenyi Biotec | Cat# 130-052-301, RRID:AB_2877061 | |

*Continued on next page*

*Continued*

| Reagent type (species) or resource | Designation | Source or reference | Identifiers | Additional information |
|---|---|---|---|---|
| Antibody | Anti-EpCAM-PE/Cy7 (Rat monoclonal, Clone G8.8) | BioLegend | Cat# 118216, RRID:AB_1236471 | FC (1:100) |
| Antibody | Anti-Ly51-Alexa Fluor 647 (Rat monoclonal, clone 6C3) | BioLegend | Cat# 108312, RRID:AB_2099613 | FC (1:250) |
| Antibody | Anti-Ly51-PE (Rat monoclonal, clone 6C3) | BioLegend | Cat# 108308, RRID:AB_313365 | FC (1:100) |
| Antibody | Ulex Europaeus Agglutinin I (UEA1)-Biotin | Vector Laboratories | Cat# B-1065, RRID:AB_2336766 | FC (1:1000) |
| Antibody | Ulex Europaeus Agglutinin I (UEA1)-DyLight 649 | Vector Laboratories | Cat# DL-1068 | FC (1:500) |
| Antibody | Anti-CD4-APC (Rat monoclonal, Clone RM4-5) | eBioscience | Cat# 17-0042-81, RRID:AB_469322 | FC (1:100) |
| Antibody | Anti-CD8α-eFluor 450 (Rat monoclonal, Clone clone 53-6.7) | Invitrogen | Cat# 48-0081-80, RRID:AB_1272235 | FC (1:200) |
| Antibody | Anti-CD25-FITC (Rat monoclonal, Clone PC61) | BioLegend | Cat# 102006, RRID:AB_312855 | FC (1:250) |

*Continued on next page*

| Reagent type (species) or resource | Designation | Source or reference | Identifiers | Additional information |
|---|---|---|---|---|
| Antibody | Anti-CD44-PE/Cy7 (Rat monoclonal, Clone IM7) | eBioscience | Cat# 25-0441-81, RRID:AB_469622 | FC (1:100) |
| Antibody | Anti-TCRβ-PE (Armenian hamster monoclonal, Clone H57-597) | BioLegend | Cat# 109207, RRID:AB_313430 | FC (1:100) |
| Antibody | Anti-TCRδ-Biotin (Armenian hamster monoclonal, Clone GL3) | BioLegend | Cat# 118103, RRID:AB_313827 | FC (1:250) |
| Antibody | Anti-Vγ5-FITC (Hamster monoclonal, Clone 536) | BD Pharmingen | Cat# 553229, RRID:AB_394721 | FC (1:100) |
| Antibody | Anti-Vγ4-FITC (Armenian hamster monoclonal, Clone UC3-10A6) | BD Pharmingen | Cat# 553226, RRID:AB_394720 | FC (1:100) |
| Antibody | Anti-Vγ1-FITC (Armenian hamster monoclonal, Clone 2.11) | BioLegend | Cat# 141103, RRID:AB_10694242 | FC (1:100) |
| Antibody | Anti-CD3ε-APC (Armenian hamster monoclonal, Clone 145-2C11) | BioLegend | Cat# 100312, RRID:AB_312677 | FC (1:20) |
| Antibody | Mouse IgG1 (Mouse monoclonal, Clone MOPC-21) | BD Pharmingnen | Cat# 554121, RRID:AB_395252 | FC (1:200) |
| Antibody | Anti-chicken IgG-Alexa Fluor 488 (Goat polyclonal) | Molecular Probes | Cat# A-11039, RRID:AB_142924 | IHC (1:500) |
| Antibody | Anti-mouse IgG-Alexa Fluor 488 (Goat polyclonal) | Invitrogen | Cat# A-11001, RRID:AB_2534069 | IHC (1:500) |
| Antibody | Anti-mouse IgG-Alexa Fluor Plus 488 (Goat polyclonal) | Invitrogen | Cat# A32723, RRID:AB_2633275 | FC (1:800) |
| Antibody | Anti-mouse IgG2a-Alexa Fluor 555 (Goat polyclonal) | Molecular Probes | Cat# A-21137, RRID:AB_2535776 | IHC (1:500) |
| Antibody | Anti-rabbit IgG-Alexa Fluor 488 (Goat polyclonal) | Invitrogen | Cat# A-11034, RRID:AB_2576217 | IHC (1:500) |

*Continued*

| Reagent type (species) or resource | Designation | Source or reference | Identifiers | Additional information |
|---|---|---|---|---|
| Antibody | Anti-rabbit IgG-Alexa Fluor 555 (Goat polyclonal) | Invitrogen | Cat# A-21429, RRID:AB_2535850 | IHC (1:500) |
| Antibody | Anti-rat IgG-Alexa Fluor 555 (Goat polyclonal) | Invitrogen | Cat# A-21434, RRID:AB_2535855 | IHC (1:500) |
| Peptide, recombinant protein | Streptavidin-Alexa Fluor 488 | Invitrogen | Cat# S32354, RRID:AB_2315383 | IHC (1:500) |
| Peptide, recombinant protein | Streptavidin-APC-eFluor 780 | Invitrogen | Cat# 47-4317-82, RRID:AB_10366688 | FC (1:80) |
| Peptide, recombinant protein | Streptavidin-PE | Invitrogen | Cat# S866 | FC (1:400) |
| Peptide, recombinant protein | DNase I | Roche | Cat# 04716728001 | (0.01%) |
| Commercial assay or kit | RNeasy Plus Micro Kit | Qiagen | Cat# 74,034 | |
| Commercial assay or kit | SMART-seq v4 Ultra Low Input RNA Kit for Sequencing | Takara Bio | Cat# 634,888 | |
| Commercial assay or kit | Nextera XT DNA Library Preparation Kit | Illumina | Cat# FC-131-1024 | |
| Commercial assay or kit | NextSeq 500/550 High Output Kit v2.5 (75 cycles) | Illumina | Cat# 20024906 | |
| Software, algorithm | GraphPad Prism | GraphPad | RRID:SCR_002798 | v7.0e |
| Software, algorithm | CLC Genomics Workbench | Qiagen | RRID:SCR_011853 | v12.0 |
| Other | poly(I:C) HMW | InvivoGen | Cat# tlrl-pic | |
| Other | TO-PRO3 | Thermo Fisher Scientific | Cat# T3605 | IHC (1:1000) |
| Other | Liberase TM | Roche | Cat# 5401127001 | (0.5 or 1 unit/ml) |

## Mice

C57BL/6 mice were obtained from SLC Japan. β5t-iCre (*Ohigashi et al., 2013*), *Ctnnb1* floxed (β-cat$^{fl}$) (*Brault et al., 2001*), *Ctnnb1* exon 3 floxed (β-cat$^{ex3fl}$) (*Harada et al., 1999*), and Rosa26-CAG-loxP-stop-loxP-tdTomato (R26R-tdTomato) (*Madisen et al., 2010*) used in this study have been described previously and genotyped accordingly. To generate conditional targeting mice, we used male mice harboring β5t-iCre allele for the mating to avoid potential germline recombination rarely observed in the offspring of female mice harboring β5t-iCre allele. TEC-specific β-catenin gain-of-function (β-cat GOF) (β5t$^{iCre/+}$;β-cat$^{ex3fl/+}$) mice were generated by intercrossing β-cat$^{ex3fl/ex3fl}$ females with β5t$^{iCre/+}$ male mice. For the lineage tracing of cells, C57BL/6 females and β-cat$^{ex3fl/ex3fl}$ females were crossed with β5t$^{iCre/+}$;R26R$^{tdTomato/tdTomato}$ males to produce β5t$^{iCre/+}$;R26R$^{tdTomato/+}$ (control) and β5t$^{iCre/+}$;β-cat$^{ex3fl/+}$; R26R$^{tdTomato/+}$ (β-cat GOF) mice, respectively. TEC-specific β-catenin loss-of-function (β-cat LOF) (β5t$^{iCre/+}$;β-cat$^{fl/fl}$) mice were generated by intercrossing β-cat$^{fl/fl}$ females with β5t$^{iCre/+}$;β-cat$^{fl/+}$ males.

Embryos were staged on the basis of standard morphological criteria; plug date was considered to be E0.5. Postnatal mice were analyzed at indicated ages up to 6 months old in an age-matched manner. Mice were housed in a 12 hr light–dark cycle in climate-controlled, pathogen-free barrier facilities. All mouse experiments were performed with consent from the Animal Experimentation Committee of the University of Tokushima (T2019-62).

## Poly(I:C) treatment

Four-week-old mice were intraperitoneally injected with 250 µg of high-molecular weight poly(I:C) (InvivoGen) twice over a 3-day interval.

## Immunohistochemistry

Embryos and thymus tissues were fixed overnight in 4% PFA/PBS, cryoprotected in 30% sucrose, embedded in optimal cutting temperature compound (Sakura Finetek), and sectioned at 10 µm thickness. Immunohistochemistry was performed on the cryosections using the following primary antibodies: rabbit anti-Foxn1 antiserum (*Itoi et al., 2006*, 1:100), rabbit anti-β5t antibody (*Murata et al.,*

*2007*, 1:200), rat anti-CCL21/6Ckine antibody (R&D systems, 1:10), eFluor 660-conjugated anti-Aire antibody (e-Bioscience, 1:20), and biotinylated anti-CD45 antibody (BioLegend, 1:100). Immunohistochemistry using monoclonal mouse anti-β-catenin (BD Transduction Laboratories, 1:200), chicken anti-Keratin 5 antibody (BioLegend, 1:500), and mouse anti-Keratin 8 antibody (BioLegend, 1:500) was performed after heat-induced antigen retrieval. For detection of antibodies, Alexa Fluor-conjugated secondary antibodies or streptavidin (Invitrogen) were used at 1:500 dilution as necessary. Nuclear counterstaining was performed using TO-PRO3 (Thermo Fisher Scientific, 1:1000). Fluorescent images were obtained and analyzed with a TCS SP8 (Leica) confocal laser scanning microscope.

## Flow cytometry and cell sorting

For flow cytometric analysis of TECs, minced thymuses from fetuses and postnatal mice were digested with 0.5 and 1 unit/ml Liberase TM (Roche) in the presence of 0.01% DNase I (Roche), respectively. Single-cell suspensions were stained for the expression of EpCAM (BioLegend, clone G8.8), CD45 (BioLegend, clone 30-F11), and Ly51 (BioLegend, clone 6C3) and for the reactivity with UEA-1 (Vector Laboratories). For the intracellular staining of β-catenin in TECs, surface-stained cells were fixed in 2% paraformaldehyde and permeabilized with 0.05% saponin, and stained with monoclonal mouse anti-β-catenin against aa. 571–781 (BD Transduction Laboratories, Clone 14, 1:100) or mouse IgG1 isotype control (BD Pharmingnen, clone MOPC-21, 1:200), followed by Alexa Fluor Plus 488-conjugated anti mouse IgG antibody (Invitrogen, 1:800).

For the isolation of TECs, CD45⁻ cells were enriched with magnetic-bead-conjugated anti-CD45 antibody (Miltenyi Biotec) before multicolor staining for flow cytometric cell sorting.

For flow cytometric analysis of thymocytes, spleen cells, and lymph node cells, cells were multicolor stained for CD4 (eBioscience, clone RM4-5), CD8α (Invitrogen, clone 53-6.7), CD25 (BioLegend, clone PC61), CD44 (eBioscience, clone IM7), TCRβ (BioLegend, clone H57-597), TCRδ (BioLegend, clone GL3), Vγ5 (BD Pharmingen, clone 536), Vγ4 (BD Pharmingen, clone UC3-10A6), and Vγ1 (BioLegend, clone 2.11).

For the analysis of DETCs, epidermal cells were prepared as described previously (*Liu et al., 2006*). Briefly, dorsal skin of 8-week-old mice was placed dermal side down in 0.25% trypsin–EDTA in PBS at 37°C for 1 hr and epidermal sheet was peeled off from dermis. Minced epidermal sheets were mechanically dissociated to make a single-cell suspension in 1xPBS containing 2% FCS. Cells were filtered and cell surface staining was performed using antibodies specific for CD3ε (eBioscience, clone145-2C11) and Vγ5 (BD Pharmingen, clone 536).

## Quantitative real-time polymerase chain reaction analysis

Total RNA was isolated using an RNeasy Plus Micro Kit (Qiagen) according to the manufacturer's instructions. Total RNA was reverse transcribed (RT) with PrimeScript Reverse Transcriptase (TaKaRa) to produce first-strand cDNA. Quantitative real-time polymerase chain reaction (PCR) was performed using TB Green Premix Taq II (TaKaRa) and a StepOnePlus Real-Time PCR System (Applied Biosystems). Values were calculated by using the comparative $C(t)$ method and normalized to mouse *Gapdh* expression. Most of the primer sets amplified products across exon/intron boundaries and the PCR products were confirmed by gel electrophoresis and melting curves. Sequences are available by request.

RNA sequencing analysis cDNA was generated from 300 isolated TECs by using a SMART-seq v4 Ultra Low Input RNA Kit for Sequencing, according to the manufacturer's protocol (Takara Bio). A sequencing library was prepared by using a Nextera XT DNA Library Preparation Kit, according to the manufacturer's protocol (Illumina). The concentration of the libraries was measured by an ABI PRISM 7500 Real-time PCR system in combination with a Power SYBR Green PCR Master Mix (Thermo Fisher Scientific). Single-end sequencing of cDNA libraries with the read length of 76 was performed with a NextSeq 550 system (Illumina) using a High Output 75-Cycle Kit (Illumina). Data were analyzed by using CLC Genomics Workbench 12 (Qiagen) with default parameters.

## Statistical analysis

Statistical analysis was performed using GraphPad Prism seven software. Statistical significance was evaluated using two-tailed unpaired Student's *t*-test with Welch's correction for unequal variances.

All values are expressed as means and standard error of the means, unless otherwise specified. The *n* numbers are indicated in figure legends.

## Acknowledgements

We thank Takashi Amagai and Manami Itoi for anti-Foxn1 antibody, Shigeo Murata for anti-β5t antibody, and Hitomi Kyuma and Yukiko Ito for technical support. We also thank all members of Takada Laboratory and Takahama Laboratory for helpful discussion.

## Additional information

### Funding

| Funder | Grant reference number | Author |
| --- | --- | --- |
| Japan Society for the Promotion of Science | 17K15740 | Sayumi Fujimori |
| Japan Society for the Promotion of Science | 17K08884 | Izumi Ohigashi |
| Takeda Science Foundation | | Izumi Ohigashi |
| Sumitomo Foundation | | Izumi Ohigashi |
| Novartis Foundation | | Izumi Ohigashi |
| Japan Society for the Promotion of Science | 24111001 | Yousuke Takahama |
| Japan Society for the Promotion of Science | 16H02630 | Yousuke Takahama |
| National Cancer Institute | Intramural Research Program | Yousuke Takahama |
| Japan Society for the Promotion of Science | 24111002 | Shinji Takada |

The funders had no role in study design, data collection, and interpretation, or the decision to submit the work for publication.

### Author contributions

Sayumi Fujimori, Conceptualization, Formal analysis, Funding acquisition, Investigation, Project administration, Validation, Visualization, Writing – original draft, Writing – review and editing; Izumi Ohigashi, Funding acquisition, Investigation, Writing – original draft, Writing – review and editing; Hayato Abe, Yosuke Matsushita, Toyomasa Katagiri, Investigation; Makoto M Taketo, Resources; Yousuke Takahama, Shinji Takada, Conceptualization, Funding acquisition, Supervision, Writing – original draft, Writing – review and editing

### Author ORCIDs

Sayumi Fujimori http://orcid.org/0000-0002-1822-1296
Izumi Ohigashi http://orcid.org/0000-0003-0017-6957
Yousuke Takahama http://orcid.org/0000-0002-4992-9174
Shinji Takada http://orcid.org/0000-0003-4125-6056

### Ethics

All mouse experiments were performed with consent from the Animal Experimentation Committee of the University of Tokushima (T2019-62).

### Decision letter and Author response

Decision letter https://doi.org/10.7554/eLife.69088.sa1
Author response https://doi.org/10.7554/eLife.69088.sa2

## Additional files

### Supplementary files
• Transparent reporting form

### Data availability
RNA sequencing data have been deposited in The DNA Data Bank of Japan (https://www.ddbj.nig.ac.jp) with accession number DRA013141.

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
