## [Editor Report]

This paper is of interest to scientists within the field of thymus development and function. The work presented builds upon previous studies that have shown that alterations of Wnt/β-catenin signaling in thymic epithelial cells (TEC) impact the normal development and or maintenance of thymic epithelial microenvironment critical for the proper development and selection of functional self-tolerant T cell repertoire. The surprise that a TEC specific loss of function of β-catenin only showed a rather minor phenotype is interesting, as are the findings that gain of function of β-catenin leads to TEC differentiation towards a keratinocyte-like lineage outcome. The author's claims are well supported by the data presented and will be of great interest to scientists and clinicians interested in understanding the signaling pathways important in thymic maintenance, as well as the development of strategies to counteract thymic involution in the aging population and cancer patients.

---

## [Decision Letter]

**Decision letter after peer review:**

Thank you for submitting your article "Fine-tuning of β-catenin in thymic epithelial cells is required for postnatal T cell development" for consideration by *eLife*. Your article has been reviewed by 3 peer reviewers, including Juan Carlos Zúñiga-Pflücker as the Reviewing Editor and Reviewer #1, and the evaluation has been overseen by Tadatsugu Taniguchi as the Senior Editor. The following individual involved in review of your submission has agreed to reveal their identity: Mark Pezzano (Reviewer #3).

Essential revisions:

1) The impact of the work would be improved by the addition of analyses that would point to a more mechanistic understanding as to the observed phenotype. These are highlighted by the characterization of Notch ligand expression in the GoF mouse, and further characterization of the terminally differentiated keratinocytes.

2) Additional analysis of the LoF mouse, either in aging or following stress/damage induced recovery would add impact to the work. Additional characterization of the cTEC subset in the LoF mice to address whether these cells have a defect in their survival/proliferation/differentiation.

3) There is a need to improve on the image analyses and cell sorting parameters.

*Reviewer #1 (Recommendations for the authors):*

The work presented by the authors beautifully illustrate the minor role that WNT signaling plays in fetal and post-natal TEC differentiation and function, which is somewhat different than what would have been expected given previous reports. Nevertheless, the authors' use of the β5t-cre is very elegant and directly addresses the role of β-catenin in TECs. Given the minor phenotype seen in the LoF experiments, it would be important to show whether non-steady state challenges and recovery would be affected in the absence of β-catenin.

1. To address a potential role for WNT in non-steady situations for TEC differentiation/function and/or recovery, the authors should consider either a radiation, steroid/inflammation, or any other stress to test whether TEC recovery is affected in the LoF mice.

The results from the GoF studies clearly point to a severe dysfunction of TECs expressing constitutively active β-catenin, highlighted by the loss of FOXN1 expression and appearance of terminally differentiated keratinocytes.

2. The lack of T cell development could be further established by showing that DLL4 expression is lost in these TECs, thus no induction of T cell development would be expected. This can be easily addressed by RT-PCR or flow cytometry of TECs. Given that Dll4 is a target of FOXN1, much like β5t and CCL25, its expression is likely totally gone, but, nonetheless, it would be good to show it.

*Reviewer #2 (Recommendations for the authors):*

The study is descriptive and needs to address in greater depth the physiological and molecular effects of stabilizing or ablating β-catenin in TECs. The authors need to provide more evidence that stabilization of β-catenin function in TECs results in their terminal differentiation to keratinocytes. The authors need to specifically assess the selective effect of loss of β-catenin function on cTECs versus mTECs. Are cTECs reduced due to loss of survival/proliferation/differentiation? How is their molecular profile affected by the loss of β-catenin? Does the selective reduction of the cTEC compartment affect survival/proliferation/differentiation of the individual DN subsets?

---

## [Author Response]

Essential revisions:1) The impact of the work would be improved by the addition of analyses that would point to a more mechanistic understanding as to the observed phenotype. These are highlighted by the characterization of Notch ligand expression in the GoF mouse, and further characterization of the terminally differentiated keratinocytes.

Regarding the characterization of Notch ligand expression, we examined DLL4 mRNA expression in TECs of β-cat GOF mice because Notch ligand DLL4 was essential for early T cell development. We found that the expression of DLL4 was markedly reduced in β-cat GOF TECs. This result agrees with the finding of no T cell development detectable in the thymus of β-cat GOF mice and is included in Figure 2B of the revised manuscript.

Regarding terminally differentiated keratinocytes, we examined the expression of additional genes relevant to cornification or corneocyte cohesion, such as keratin 1 (*Krt1*), keratin 10 (*Krt10*), and desmoglein-1 α (*Dsg1a*), which are associated with the terminal differentiation of keratinocytes. We found that the expression of *Dsg1a* was elevated in β-cat GOF TECs in a manner similar to upregulated involucrin (*Ivl*) and loricrin (*Lor*) expression, as described in the original manuscript. However, we also found that *Krt1* and *Krt10* were not detectable in those β-cat GOF TECs. Based on these results, we concluded that β-cat GOF TECs exhibited aberrant differentiation into epithelial cells that partially shared gene expression profiles with the terminally differentiated keratinocytes. We have included these additional results in Figure 2B of the revised manuscript. We have also described in the Discussion of the revised manuscript that β-cat GOF TECs were not exactly equivalent to the terminally differentiated keratinocytes because *Krt1* and *Krt10* were not detectable in β-cat GOF TECs.

Thank you very much for the kind suggestions to help improve the manuscript.

2) Additional analysis of the LoF mouse, either in aging or following stress/damage induced recovery would add impact to the work. Additional characterization of the cTEC subset in the LoF mice to address whether these cells have a defect in their survival/proliferation/differentiation.

According to the suggestion, we performed the following new experiments on β-cat LOF mice: (1) RNA sequencing analysis of cTECs and mTECs at 2 weeks old, (2) analysis of the thymus following polyinosinic:polycytidylic acid (poly(I:C))-mediated stress-induced injury at 4 weeks old, and (3) analysis of the thymus at 6 months old.

(1) By RNA sequencing analysis, we detected 92 genes that were significantly altered in β-cat LOF cTECs with the false discovery rate (FDR) adjusted p-value of less than 0.05; there were 11 downregulated genes and 81 upregulated genes compared with control cTECs. We also detected 91 genes that were significantly altered in β-cat LOF mTECs; there were 53 downregulated genes and 38 upregulated genes compared with control mTECs. Thus, only less than 100 genes were significantly altered in cTECs and mTECs due to LOF of β-catenin, in agreement with the finding of no remarkable alteration in the expression of many functionally relevant molecules in cTECs and mTECs by quantitative RT-PCR analysis, as shown in the original manuscript. Nonetheless, we found that a vast majority of the genes that were significantly affected by LOF of β-catenin were different between cTECs and mTECs except *Adh1* and *Pglyrp1*. *Adh1* was upregulated in β-cat LOF cTECs and downregulated in β-cat LOF mTECs, whereas *Pglyrp1* was upregulated by β-cat LOF in both cTECs and mTECs. Notably, *Cdkn1a* was elevated specifically in cTECs but not in mTECs in β-cat LOF mice. The expression of *Cdkn1a*, which encodes cyclin-dependent kinase (CDK) inhibitor p21, is linked to β-catenin activity. The upregulation of *Cdkn1a* in β-cat LOF cTECs compared with control cTECs was confirmed by quantitative RT-PCR analysis. We noticed no remarkable difference in other CDK family genes, including *Cnnb1*, *Cnnb2*, and *Cnnd1*, in cTECs and mTECs from β-cat LOF mice. These results suggest that the upregulation of *Cdkn1a* may contribute to the reduction in the number of cTECs by LOF of β-catenin. These new results are shown in new Figure 8 and explained as well as discussed in the revised manuscript.

(2) We also examined how stress-induced injury would affect the thymus in β-cat LOF mice. To do so, β-cat LOF mice were treated with polyinosinic:polycytidylic acid (poly(I:C)), a synthetic analog of double-stranded RNA, which caused injury in TECs. We found that both control and β-cat LOF mice showed transient thymic involution that was evident from the loss of thymus weight and total thymic cell number four days after the first poly(I:C) administration. Subsequently, however, the involuted thymus recovered to its pretreatment size by day 18 and day 25 in both control and β-cat LOF mice, indicating that the small thymus in postnatal β-cat LOF mice retained the capability to recover from the poly(I:C)-mediated thymic injury. Indeed, the cTECs and mTECs recovered to their numbers before the treatment. Interestingly, we noticed a delay in the recovery of the thymus in β-cat LOF mice compared with that in control mice. The weight and the cell number of the thymus were still significantly lower on day 11 than day 0 in β-cat LOF mice, unlike the recovery to the pretreatment levels on day 11 in control mice. Accordingly, the number of DP thymocytes remained slightly but significantly reduced even on day 25 in β-cat LOF mice. These results indicate that LOF of β-catenin in TECs delays the recovery of the thymus from the poly(I:C)-induced thymic involution, but does not fully abolish the capability of the thymus to recover from the poly(I:C)-induced thymic injury. These new results are shown in new Figure 11 and explained in the revised manuscript.

(3) Furthermore, we examined the phenotype of the thymus in β-cat LOF mice at 6 months old (6 mo), in which the involution of the thymus was apparent. The weight of the thymus decreased in control mice at 6 mo in comparison with those at 4 weeks old (4 wo) and 8 wo. Similarly, the thymus weight decreased in β-cat LOF mice at 6 mo compared with that at 4 wo. Accordingly, the thymus at 6 mo was still smaller in β-cat LOF mice than control mice. The number of thymocytes was reduced upon the thymic involution in β-cat LOF mice, although the difference in thymocyte number became insignificant between control mice and β-cat LOF mice at 6 mo. The number of cTECs in β-cat LOF mice at 6 wo remained smaller than the control number, whereas the number of mTECs was equivalent to that of control at 6 mo. There were no apparent differences in the corticomedullary architecture of the thymus between control mice and β-cat LOF mice at 6 mo. These results indicate that age-associated thymic involution is detectable in β-cat LOF mice. These new results are shown in new Figure 10 and explained in the revised manuscript.

Thank you very much for giving us the opportunity to perform additional analyses of the thymus in β-cat LOF mice. The newly obtained results have given us deeper insight into the defects in TECs, particularly in cTECs, in β-cat LOF mice, which have been additionally described in the revised manuscript.

3) There is a need to improve on the image analyses and cell sorting parameters.

We have added individual single-color immunohistological images in new Figure 1—figure supplement 1 and Figure 2—figure supplement 1A in the revised manuscript. We have also included flow cytometric cell-sorting gates and post-sorting purities of TECs, cTECs, and mTECs in new Figure 2—figure supplement 1B and Figure 7—figure supplement 1 in the revised manuscript. Thank you very much for helping us improve the manuscript.

Reviewer #1 (Recommendations for the authors):The work presented by the authors beautifully illustrate the minor role that WNT signaling plays in fetal and post-natal TEC differentiation and function, which is somewhat different than what would have been expected given previous reports. Nevertheless, the authors' use of the β5t-cre is very elegant and directly addresses the role of β-catenin in TECs. Given the minor phenotype seen in the LoF experiments, it would be important to show whether non-steady state challenges and recovery would be affected in the absence of β-catenin.1. To address a potential role for WNT in non-steady situations for TEC differentiation/function and/or recovery, the authors should consider either a radiation, steroid/inflammation, or any other stress to test whether TEC recovery is affected in the LoF mice.The results from the GoF studies clearly point to a severe dysfunction of TECs expressing constitutively active β-catenin, highlighted by the loss of FOXN1 expression and appearance of terminally differentiated keratinocytes.

According to the reviewer’s suggestion, we have also examined how stress-induced injury would affect the thymus in β-cat LOF mice. To do so, β-cat LOF mice were treated with polyinosinic:polycytidylic acid (poly(I:C)), a synthetic analog of double-stranded RNA, which caused injury in TECs. We found that both control and β-cat LOF mice showed transient thymic involution that was evident from the loss of thymus weight and total thymic cell number four days after the first poly(I:C) administration. Subsequently, however, the involuted thymus recovered to its pretreatment size by day 18 and day 25 in both control and β-cat LOF mice, indicating that the small thymus in postnatal β-cat LOF mice retained the capability to recover from the poly(I:C)-mediated thymic injury. Indeed, the cTECs and mTECs recovered to their numbers before the treatment. Interestingly, we noticed a delay in the recovery of the thymus in β-cat LOF mice compared with that in control mice. The weight and the cell number of the thymus were still significantly lower on day 11 than day 0 in β-cat LOF mice, unlike the recovery to the pretreatment levels on day 11 in control mice. Accordingly, the number of DP thymocytes remained slightly but significantly reduced even on day 25 in β-cat LOF mice. These results indicate that LOF of β-catenin in TECs delays the recovery of the thymus from the poly(I:C)-induced thymic involution, but does not fully abolish the capability of the thymus to recover from the poly(I:C)-induced thymic injury. These new results are shown in new Figure 11 and explained in the revised manuscript. Thank you very much for giving us the opportunity to provide deeper insight into the defects in TECs, particularly in cTECs, in β-cat LOF mice.

2. The lack of T cell development could be further established by showing that DLL4 expression is lost in these TECs, thus no induction of T cell development would be expected. This can be easily addressed by RT-PCR or flow cytometry of TECs. Given that Dll4 is a target of FOXN1, much like β5t and CCL25, its expression is likely totally gone, but, nonetheless, it would be good to show it.

We have examined DLL4 mRNA expression in TECs of β-cat GOF mice because Notch ligand DLL4 is essential for early T cell development. We found that the expression of DLL4 was markedly reduced in β-cat GOF TECs. These results agree with the finding of no T cell development detectable in the thymus of β-cat GOF mice, and are included in Figure 2B of the revised manuscript. Thank you very much for your kind suggestions to help improve the manuscript.

Reviewer #2 (Recommendations for the authors):The study is descriptive and needs to address in greater depth the physiological and molecular effects of stabilizing or ablating β-catenin in TECs. The authors need to provide more evidence that stabilization of β-catenin function in TECs results in their terminal differentiation to keratinocytes.

According to the suggestion, we examined the expression of additional genes relevant to cornification or corneocyte cohesion, such as keratin 1 (*Krt1*), keratin 10 (*Krt10*), and desmoglein-1 α (*Dsg1a*), which are associated with the terminal differentiation of keratinocytes. We found that the expression of *Dsg1a* was elevated in β-cat GOF TECs in a manner similar to upregulated involucrin (*Ivl*) and loricrin (*Lor*) expression, as described in the original manuscript. However, we also found that *Krt1* and *Krt10* were not detectable in those β-cat GOF TECs. Based on these results, we concluded that β-cat GOF TECs exhibited aberrant differentiation into epithelial cells that partially shared gene expression profiles with the terminally differentiated keratinocytes. We have included these additional results in Figure 2B of the revised manuscript. We have also described in the Discussion of the revised manuscript that β-cat GOF TECs were not exactly equivalent to the terminally differentiated keratinocytes because *Krt1* and *Krt10* were not detectable in β-cat GOF TECs. Thank you very much for the suggestion to help improve the manuscript.

The authors need to specifically assess the selective effect of loss of β-catenin function on cTECs versus mTECs. Are cTECs reduced due to loss of survival/proliferation/differentiation? How is their molecular profile affected by the loss of β-catenin?

Following the reviewer’s suggestion, we performed RNA sequencing analysis of cTECs and mTECs in β-cat LOF mice. We detected 92 genes that were significantly altered in β-cat LOF cTECs with the FDR adjusted p-value of less than 0.05; there were 11 downregulated genes and 81 upregulated genes compared with control cTECs. We also detected 91 genes that were significantly altered in β-cat LOF mTECs; there were 53 downregulated genes and 38 upregulated genes compared with control mTECs. Thus, only less than 100 genes were significantly altered in cTECs and mTECs due to LOF of β-catenin, in agreement with the finding of no remarkable alteration in the expression of many functionally relevant molecules in cTECs and mTECs by quantitative RT-PCR analysis, as shown in the original manuscript. Nonetheless, we found that a vast majority of the genes that were significantly affected by LOF of β-catenin were different between cTECs and mTECs except *Adh1* and *Pglyrp1*. *Adh1* was upregulated in β-cat LOF cTECs and downregulated in β-cat LOF mTECs, whereas *Pglyrp1* was upregulated by β-cat LOF in both cTECs and mTECs. Notably, *Cdkn1a* was elevated specifically in cTECs but not in mTECs in β-cat LOF mice. The expression of *Cdkn1a*, which encodes cyclin-dependent kinase (CDK) inhibitor p21, is linked to β-catenin activity. The upregulation of *Cdkn1a* in β-cat LOF cTECs compared with control cTECs was confirmed by quantitative RT-PCR analysis. We noticed no remarkable difference in other CDK family genes, including *Cnnb1*, *Cnnb2*, and *Cnnd1*, in cTECs and mTECs from β-cat LOF mice. These results suggest that the upregulation of *Cdkn1a* may contribute to the selective reduction in the number of cTECs by LOF of β-catenin. These new results are shown in new Figure 8 and explained as well as discussed in the revised manuscript. Thank you so much for your comments to help improve the manuscript.

Does the selective reduction of the cTEC compartment affect survival/proliferation/differentiation of the individual DN subsets?

According to the reviewer’s suggestion, we examined the DN subsets in β-cat LOF mice and found that the frequency of DN subsets defined by CD44 and CD25 was unchanged, although the number of all four DN subsets was significantly reduced in β-cat LOF mice. These new results are shown in new Figure 9 and explained in the revised manuscript. Thank you so much for your comments to help improve the manuscript.